# Spectral Augmentation for Self-Supervised Learning on Graphs

**Lu Lin**[†§]**, Jinghui Chen**[†]**, Hongning Wang**[§]
[†]The Pennsylvania State University, [§]University of Virginia
`{lulin,jzc5917}@psu.edu, hw5x@virginia.edu`

## Abstract

Graph contrastive learning (GCL), as an emerging self-supervised learning technique on graphs, aims to learn representations via instance discrimination. Its performance heavily relies on graph augmentation to reflect invariant patterns that are robust to small perturbations; yet it still remains unclear about *what graph invariance GCL should capture*. Recent studies mainly perform topology augmentations in a uniformly random manner in the spatial domain, ignoring its influence on the intrinsic structural properties embedded in the spectral domain. In this work, we aim to find a principled way for topology augmentations by exploring the invariance of graphs from the spectral perspective. We develop spectral augmentation which guides topology augmentations by maximizing the spectral change. Extensive experiments on both graph and node classification tasks demonstrate the effectiveness of our method in unsupervised learning, as well as the generalization capability in transfer learning and the robustness property under adversarial attacks. Our study sheds light on a general principle for graph topology augmentation.

## 1 Introduction

Graph neural networks (GNNs) (Kipf & Welling, 2017; Veličković et al., 2018; Xu et al., 2019) have advanced graph representation learning in a (semi-)supervised manner, yet it requires supervised labels and may fail to generalize (Rong et al., 2020). To obtain more generalizable and transferable representations, the self-supervised learning (SSL) paradigm emerges which enables GNNs to learn from pretext tasks constructed on unlabeled graph data (Hu et al., 2020c;b; You et al., 2020b; Jin et al., 2020a). As a state-of-the-art SSL technique, graph contrastive learning (GCL) has attracted the most attention due to its remarkable empirical performance (Velickovic et al., 2019; Zhu et al., 2020; Hassani & Khasahmadi, 2020; You et al., 2021; Suresh et al., 2021; Thakoor et al., 2021).

A typical GCL method works by creating augmented views of the input graph and learning representations by contrasting related graph objects against unrelated ones. Different contrastive objects are studied on graphs, such as node-node (Zhu et al., 2020; 2021; Peng et al., 2020), node-(sub)graph (Veličković et al., 2019; Hassani & Khasahmadi, 2020; Sun et al., 2019) and graph-graph (Bielak et al., 2021; Thakoor et al., 2021; Suresh et al., 2021) contrastive pairs. The goal of GCL is to capture graph invariance by maximizing the congruence between node or graph representations in augmented views. This makes graph augmentation one of the most critical designs in GCL, as it determines the effectiveness of the contrastive objective. However, despite that various GCL methods have been proposed, it remains a mystery about *what graph invariance GCL should capture.*

Unlike images, which can be augmented to naturally highlight the main subject from the background, it is less obvious to design the most effective graph augmentation due to the complicated topology structure of diverse nature in different graphs (e.g., citation networks (Sen et al., 2008), social networks (Morris et al., 2020), chemical and biomedical molecules (Li et al., 2021; Hu et al., 2020b)), as discussed in the survey (Ding et al., 2022). We argue that an ideal GCL encoder should preserve *structural invariance*, and an effective augmentation focuses on perturbing edges leading to large changes in structural properties; and by maximizing the congruence across the resulting views, information about sensitive or friable structures will be minimized in the learned representations.

Most existing works perform topology augmentations in a uniformly random manner (Zhu et al., 2020; Thakoor et al., 2021), which achieves a certain level of empirical success, but is far from optimal:

recent studies show that perturbations on different edges post *unequal* influence on the structural properties (Entezari et al., 2020; Chang et al., 2021a) while the uniformly random edge perturbation ignores such differences. Such a discrepancy suggests an opportunity to improve the common uniform augmentation by considering structural properties to better capture structural invariance. Since graph spectrum summarizes many important structural properties (Chung & Graham, 1997), we propose to preserve *spectral invariance* as a proxy of structural invariance, which refers to the invariance of the encoder's output to perturbations on edges that cause large changes on the graph spectrum.

To realize the spectral invariance, we focus on designing a principled augmentation method from the perspective of graph spectrum, termed ***SPectral AugmentatioN (SPAN)***. Specifically, we search for topology augmentations that achieve the largest disturbance on graph spectrum. By identifying sensitive edges whose perturbation leads to a large spectral difference, SPAN allows the GNN encoder to focus on robust spectrtal components (which can be hardly affected by small edge perturbations) and to reduce its dependency on instable ones (which can be easily affected by the perturbation). Therefore, the learned encoder captures the minimally information about the graph (Tishby et al., 2000; Tian et al., 2020) for downstream tasks.

We provide an instantiation of GCL on top of the proposed augmentation method SPAN, which can also be easily paired with different GCL paradigms as it only requires a one-time pre-computation of the edge perturbation probability. The effectiveness of SPAN is extensively evaluated on various benchmark datasets, which cover commonly seen graph learning tasks such as node classification, graph classification and regression. The applicability of SPAN is tested under various settings including unsupervised learning, transfer learning and adversarial learning setting. In general, SPAN achieves remarkable performance gains compared to the state-of-the-art baselines. Our study can potentially open up new ways for topology augmentation from the perspective of graph spectrum.

## 2 RELATED WORKS

**Graph Contrastive Learning (GCL)** leverages the InfoMax principle (Hjelm et al., 2018) to maximize the correspondence between related objects on the graph such that invariant property across objects is captured. Depending on how the positive objects are defined, one line of work treats different parts of a graph as positive pairs, while constructing negative examples from a corrupted graph (Hu et al., 2020b; Jiao et al., 2020; Veličković et al., 2019; Peng et al., 2020; Sun et al., 2019). In such works, contrastive pairs are defined as nodes (Veličković et al., 2019) or substructures (Sun et al., 2019) v.s. the entire graph, and the input graph v.s. reconstructed graph (Peng et al., 2020). The other line of works exploit *graph augmentation* to generate multiple views, which enable more flexible contrastive pairs (Thakoor et al., 2021; Bielak et al., 2021; Suresh et al., 2021; You et al., 2021; Feng et al., 2022; Ding et al., 2022). By generating augmented views, the GNN model is encouraged to encode crucial graph information invariant to different views. In this work, we focus on topology augmentation. As a parallel effort in self-supervised learning, augmentation-free techniques (Lee et al., 2022; Wang et al., 2022) avoid augmentation but require special treatments (e.g., kNN search or clustering) to obtain positive and negative pairs, which is out scope of this work.

**Graph Topology Augmentation.** The most widely adopted topology augmentation is the edge perturbation following *uniform distribution* (Zhu et al., 2020; Thakoor et al., 2021; Bielak et al., 2021; You et al., 2020a). The underlying assumption is that each edge is equally important to the property of the input graph. However, a recent study shows that edge perturbations do not post equal influence to the graph spectrum (Chang et al., 2021a) which summarizes a graph's structural property. To better preserve graph property that has been ignored by uniform perturbations, *domain knowledge* from network science is leveraged by considering the importance of edges measured via node centrality (Zhu et al., 2021), the global diffusion matrix (Hassani & Khasahmadi, 2020), and the random-walk based context graph (Qiu et al., 2020). While these works consider ad-hoc heuristics, our method targets at the graph spectrum, which comprehensively summarizes global graph properties and plays a crucial role in the spectral filter of GNNs. To capture minimally sufficient information from the graph and remove redundancy that could compromise downstream performance, *adversarial training* strategy is paired with GCL for graph augmentation Suresh et al. (2021); You et al. (2021); Feng et al. (2022), following the information bottleneck (IB) (Tishby et al., 2000) and InfoMin principle (Tian et al., 2020). While the adversarial augmentation method requires frequent back-propagation during training, our method realizes a similar principle with a simpler but effective augmentation by maximizing the spectral difference of views with only one-time pre-computation.

## 3 PRELIMINARIES

**Notations.** We focus on connected undirected graphs $G = (\mathbf{X}, \mathbf{A})$ with $n$ nodes and $m$ edges, where $\mathbf{X} \in \mathbb{R}^{n \times d}$ describes node features, and $\mathbf{A} \in \mathbb{R}^{n \times n}$ denotes its adjacency matrix such that $A_{ij} = 1$ if an edge exists between node $i$ and $j$, otherwise $A_{ij} = 0$. The unnormalized Laplacian matrix of the graph is defined as $\mathbf{L}_{\text{u}} = \mathbf{D} - \mathbf{A}$, where $\mathbf{D} = \text{diag}(\mathbf{A}\mathbf{1}_n)$ is the diagonal degree matrix with entry $D_{ii} = \sum_{i=1}^{n} A_{ij}$ and $\mathbf{1}_n$ being an all-one vector with dimension $n$. The normalized Laplacian matrix is further defined as $\mathbf{L}_{\text{norm}} = \text{Lap}(\mathbf{A}) = \mathbf{I}_n - \mathbf{D}^{-1/2} \mathbf{A} \mathbf{D}^{-1/2}$, where $\mathbf{I}_n$ is an $n \times n$ identity matrix.

**Graph Spectrum.** Graph representation learning can be viewed as graph signal processing (GSP). The graph shift operator (GSO) in GSP commonly adopts the normalized Laplacian matrix $\mathbf{L}_{\text{norm}}$ and admits an eigendecomposition as $\mathbf{L}_{\text{norm}} = \mathbf{U} \mathbf{\Lambda} \mathbf{U}^\top$. The diagonal matrix $\mathbf{\Lambda} = \text{eig}(\mathbf{L}_{\text{norm}}) = \text{diag}(\lambda_1, \ldots, \lambda_n)$ consists of the real eigenvalues which are known as *graph spectrum*, and the corresponding $\mathbf{U} = [\mathbf{u}_1, \ldots, \mathbf{u}_n] \in \mathbb{R}^{n \times n}$ collecting the orthonormal eigenvectors are the *spectral bases*. Graph spectrum plays a significant role in analyzing and modeling graphs, as discussed in Appendix A. On one hand, it comprehensively summarizes important graph structural properties, including connectivity (Chung & Graham, 1997), clusterability (Lee et al., 2014) and etc. On the other hand, as the essence of GNNs, *spectral filter* is defined on the graph spectrum to manipulate graph signals in various ways, such as smoothing and denoising (Schaub & Segarra, 2018).

**Graph Representation Learning.** Given a graph $G \in \mathcal{G}$, the goal of node representation learning is to train an encoder $f_\theta : \mathcal{G} \to \mathbb{R}^{n \times d'}$, such that $f_\theta(G)$ produces a low-dimensional vector for each node in $G$ which can be served in downstream tasks, such as node classification. One can further obtain a graph representation by pooling the set of node representations via a readout function $g_\phi : \mathbb{R}^{n \times d'} \to \mathbb{R}^{d'}$, such that $g_\phi(f_\theta(G))$ outputs a low-dimensional vector for graph $G$ which can be used in graph-level tasks. We use $\Theta$ to denote all the model parameters.

**Graph Contrastive Learning by Topology Augmentation.** GCL methods generally apply graph augmentation to perturb the input graph and decrease the amount of information inherited from the original graph; then they leverage the InfoMax principle (Hjelm et al., 2018) over the perturbed graph views such that an encoder is trained to capture the remaining information. Given a graph $G \in \mathcal{G}$ with adjacency matrix $\mathbf{A}$, we denote a *topology augmentation scheme* as $T(\mathbf{A})$ and a sampled augmented view as $t(\mathbf{A}) \sim T(\mathbf{A})$. GCL with two-branch augmentation can be formulated as follows:

$$\text{GCL} : \min_{\Theta} \mathcal{L}_{\text{GCL}}(t_1(\mathbf{A}), t_2(\mathbf{A}), \Theta), \text{ s.t. } t_i(\mathbf{A}) \sim T_i(\mathbf{A}), i \in \{1, 2\} \tag{1}$$

where the contrastive loss $\mathcal{L}_{\text{GCL}}$ measures the disagreement between representations from contrastive positive pairs. The topology augmentation scheme determines a distribution from which perturbed graphs are sampled in augmented views, and its detailed formulation will be presented in Section 5.

## 4 STRUCTURAL INVARIANCE MATTERS IN GCL

**Structural Invariance.** Despite that multiple topology augmentation methods have been proposed, little effort is made to answer a fundamental question: *what invariance information should GCL capture?* We argue that an ideal GCL encoder should preserve *structural invariance*, which is defined as the invariance of the encoder's output when perturbing a constrained number of edges that cause large changes to the structural properties of the input graph:

$$\mathcal{L}_{\text{GCL}}(\mathbf{A}, t(\mathbf{A}), \Theta) \leq \sigma, \text{s.t. } t(\mathbf{A}) = \text{argmax}_{\|\mathbf{A} - t(\mathbf{A})\|_1 \leq \epsilon} \mathcal{D}(p(\mathbf{A}), p(t(\mathbf{A}))) \tag{2}$$

where $\mathcal{D}(\cdot, \cdot)$ is a distance metric, and $p(\cdot)$ can be defined as a vector-valued function of the graph's structural properties, such as the diameter of the graph, whether the graph is connectivity, the clustering coefficient, etc. One may focus on particular properties when designing the function $p(\cdot)$. To realize structural invariance via GCL, effective augmentation should focus on sensitive edges whose perturbation can easily cause large property change. Then by minimizing the contrastive loss $\mathcal{L}_{\text{GCL}}$, the edges causing structural instability are regarded as noise, and the information related to these edges will be ignored by the encoder, inspired by the InfoMin principle (Tian et al., 2020).

Capturing structural invariance requires the topology augmentation to identify the sensitivity of edges to the structural properties, while the augmentation with uniformly random edge perturbation fails to realize this. The discrepancy motivates the following pre-analysis to demonstrate a potential opportunity of improvement upon uniform perturbation.

**Pre-analysis.** We apply GRACE (Zhu et al., 2020) on the Cora dataset, and compare the original uniformly random augmentation in GRACE with a simple heuristic based on the clustering property of nodes. We take the clustering property as an example to demonstrate our insight, as it is an intuitive manifestation among many structural properties. To be more specific, we compare two topology augmentation heuristics: 1) *uniform* augmentation that removes edges uniformly random; 2) *clustered* augmentation that flips edges between different node clusters with a larger probability, which can cause large changes to the

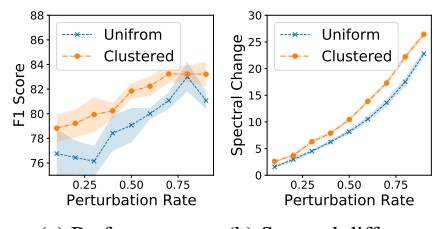

(a) Performance   (b) Spectral difference

Figure 1: Comparison between *uniform* and *clustered* augmentations.

clustering property of the graph suggested by recent studies (Chiplunkar et al., 2018; Lin et al., 2021). We use the original graph in one augmentation branch, and compare GCL performance under the uniform or the clustered augmentation in the other branch. More experiment details are provided in Appendix C.1. The downstream node classification performance evaluated by *F1 score* is compared in Figure 1a. We can clearly observe that under the same perturbation budget indicated by the x-axis, the cluster-based strategy achieves better performance on the downstream task. The performance gap between these two simple strategies suggests a distinct opportunity to improve over the uniformly random augmentation by considering structural invariance.

**From Structural Invariance to Spectral Invariance.** Directly capturing structural invariance for GCL is nontrivial, as it requires simultaneous characterization of multiple structural properties. Fortunately, recent studies show that *graph spectrum* is a comprehensive summary of many graph structural properties (Chung & Graham, 1997; Lee et al., 2014), including clustering, connectivity and etc. For example, the change on clustering property (by perturbing inter-cluster edges) can be reflected on the graph spectrum: 1) empirically, Figure 1b compares the *spectral change* (measured by the $L_2$ distance of graph spectrum between the original and the augmented graph), and we can observe a larger spectral change for the cluster-based augmentation; 2) theoretically, we can also prove that the edge flips between different clusters causing larger clustering change result in larger spectral change, as discussed in Appendix D.6. Therefore, we can use graph spectrum as a ladder to capture structural properties, and propose *spectral invariance* as a proxy of structural invariance.

Spectral invariance requires the encoder's output to stay similar when perturbing edges that cause large changes to the graph spectrum. Formally, it is defined as:

$$\mathcal{L}_{\text{GCL}}(\mathbf{A}, t(\mathbf{A}), \Theta) \leq \sigma, \text{s.t. } t(\mathbf{A}) = \operatorname*{argmax}_{\|\mathbf{A}-t(\mathbf{A})\|_1 \leq \epsilon} \mathcal{D}(\text{eig}(\text{Lap}(\mathbf{A})), \text{eig}(\text{Lap}(t(\mathbf{A})))) \quad (3)$$

where the distance between two graph structures is measured on the graph spectrum. An effective topology augmentation should pay more attention to sensitive edges that introduce large disturbances to the graph spectrum, whose influence should then be eliminated by contrastive learning. Unlike the proof-of-concept cluster-based heuristic, we aim on designing a principled augmentation by directly maximizing the spectral change.

## 5 SPECTRAL AUGMENTATION ON GRAPHS

In this section, we introduce our **SP**ectral **A**ugmentatio**N** (SPAN) on graph topology to preserve spectral invaraince in GCL. We first define the edge perturbation based topology augmentation scheme determined by a Bernoulli *probability matrix*. Based on that, we formulate our augmentation principle as a spectral change maximization problem.

**Edge Perturbation Based Augmentation Scheme.** We focus on topology augmentation using edge perturbation. Following the GCL formulation in Eq. 1, we define topology augmentation $T(\mathbf{A})$ as a Bernoulli distribution $\mathcal{B}(\Delta_{ij})$ for each entry $A_{ij}$. All Bernoulli parameters constitute a *probability matrix* $\mathbf{\Delta} \in [0, 1]^{n \times n}$. We can sample an edge perturbation matrix $\mathbf{E} \in \{0, 1\}^{n \times n}$, where $E_{ij} \sim \mathcal{B}(\Delta_{ij})$ indicates whether to flip the edge between node $i$ and $j$, and the edge is flipped if $E_{ij} = 1$ otherwise remaining unchanged. A sampled augmented graph is then obtained via:

$$t(\mathbf{A}) = \mathbf{A} + \mathbf{C} \circ \mathbf{E}, \ \mathbf{C} = \bar{\mathbf{A}} - \mathbf{A} \quad (4)$$

where $\bar{\mathbf{A}}$ is the complement matrix of the adjacency matrix $\mathbf{A}$, calculated by $\bar{\mathbf{A}} = \mathbf{1}_n \mathbf{1}_n^\top - \mathbf{I}_n - \mathbf{A}$, with $(\mathbf{1}_n \mathbf{1}_n^\top - \mathbf{I}_n)$ denoting the fully-connected graph without self-loops. Therefore, $\mathbf{C} = \bar{\mathbf{A}} - \mathbf{A} \in \{-1, 1\}^{n \times n}$ denotes legitimate edge adding or removing operations for each node pair: edge adding

between node $i$ and $j$ is allowed if $C_{ij} = 1$, and edge removing is allowed if $C_{ij} = -1$. Taking the Hadamard product $\mathbf{C} \circ \mathbf{E}$ finally gives valid edge perturbations to the graph.

Since $\mathbf{E}$ is a matrix of random variables following Bernoulli distributions, the *expectation* of sampled augmented graphs in Eq. 4 is $\mathbb{E}[t(\mathbf{A})] = \mathbf{A} + \mathbf{C} \circ \mathbf{\Delta}$. Therefore, the design of $\mathbf{\Delta}$ determines the topology augmentation scheme. Taking uniformly random edge removal as an example, the entry $\Delta_{ij}$ is set as a fixed dropout ratio if $C_{ij} = -1$; and $0$ otherwise. While we focus on edge perturbation, an extension of the proposed principle to node dropping augmentation is discussed in Appendix E.

**Spectral Change Maximization.** Motivated by our observation in Section 4, instead of setting fixed values for $\mathbf{\Delta}$ as in uniform perturbation, we propose to optimize it guided by graph spectrum. Specifically, we aim to search for $\mathbf{\Delta}$ that in expectation maximizes the spectral difference between the original graph and the augmented graph. Note that while the perturbations are sampled from $\mathbf{\Delta}$ via the resulting Bernoulli distributions, $\mathbf{\Delta}$ on all edges is jointly optimized. Denoting the normalized Laplacian matrix of $\mathbf{A}$ as $\text{Lap}(\mathbf{A})$, the graph spectrum can be calculated by $\mathbf{\Lambda} = \text{eig}(\text{Lap}(\mathbf{A}))$. We formulate the following problem to search for the desired $\mathbf{\Delta}$ in a single augmentation branch:

$$\text{Single-way scheme SPAN}_{\text{single}}: \quad \max_{\mathbf{\Delta} \in \mathcal{S}} \|\text{eig}(\text{Lap}(\mathbf{A} + \mathbf{C} \circ \mathbf{\Delta})) - \text{eig}(\text{Lap}(\mathbf{A}))\|_2^2 \tag{5}$$

where $\mathcal{S} = \{\mathbf{s} | \mathbf{s} \in [0,1]^{n \times n}, \|\mathbf{s}\|_1 \leq \epsilon\}$ and $\epsilon$ controls the perturbation strength. By solving Eq. 5, we obtain the optimal Bernoulli probability matrix $\mathbf{\Delta}^*$, from which augmented views are sampled that in expectation differ the most from the original graph in graph spectrum. Eq. 5 only provides one augmented view; to further introduce flexibility for a two-branch augmentation framework and enlarge the spectral difference between the resulting two views, we extend Eq. 5 as follows:

$$\text{Double-way SPAN}_{\text{double}}: \quad \max_{\mathbf{\Delta}_1, \mathbf{\Delta}_2 \in \mathcal{S}} \|\text{eig}(\text{Lap}(\mathbf{A} + \mathbf{C} \circ \mathbf{\Delta}_1)) - \text{eig}(\text{Lap}(\mathbf{A} + \mathbf{C} \circ \mathbf{\Delta}_2))\|_2^2 \tag{6}$$

where $\mathbf{\Delta}_i$ is the Bernoulli probability matrix for augmentation branch $i$'s scheme $T_i(\mathbf{A})$ in Eq. 1. Note that Eq. 5 is a special case of Eq. 6 when setting $\mathbf{\Delta}_2 = \mathbf{0}$. Eq. 6 gives better flexibility yet also makes the optimization problem harder to solve; thus based on triangle inequality, we maximize its lower bound instead: $\max_{\mathbf{\Delta}_1, \mathbf{\Delta}_2 \in \mathcal{S}} \|\text{eig}(\text{Lap}(\mathbf{A} + \mathbf{C} \circ \mathbf{\Delta}_1))\|_2^2 - \|\text{eig}(\text{Lap}(\mathbf{A} + \mathbf{C} \circ \mathbf{\Delta}_2))\|_2^2$. Therefore, $\mathbf{\Delta}_1$ and $\mathbf{\Delta}_2$ can be independently optimized towards opposite directions: maximizing the spectral norm in one branch, while minimizing it in the other, which leads to the final objective:

$$\text{Opposite-direction scheme SPAN}_{\text{opposite}}: \quad \max_{\mathbf{\Delta}_1 \in \mathcal{S}} \mathcal{L}_{\text{GS}}(\mathbf{\Delta}_1), \quad \text{and} \quad \min_{\mathbf{\Delta}_2 \in \mathcal{S}} \mathcal{L}_{\text{GS}}(\mathbf{\Delta}_2) \tag{7}$$

where $\mathcal{L}_{\text{GS}}(\mathbf{\Delta}) = \|\text{eig}(\text{Lap}(\mathbf{A} + \mathbf{C} \circ \mathbf{\Delta}))\|_2^2$ measures the Graph Spectral norm under augmentation scheme with $\mathbf{\Delta}$. For scheme $T_1(\mathbf{A})$, $\mathbf{\Delta}_1$ produces views that overall have larger spectral norm than the original graph, while for $T_2(\mathbf{A})$, $\mathbf{\Delta}_2$ produces views with smaller spectrum. We can understand them as setting a spectral boundary for the input graph such that the encoder is trained to capture information that is essential and robust regarding perturbations within this region.

**Optimizing $\mathbf{\Delta}_1$ and $\mathbf{\Delta}_2$.** Eq. 7 can be solved via projected gradient descent (for $\mathbf{\Delta}_2$) or ascent (for $\mathbf{\Delta}_1$). Taking $\mathbf{\Delta}_2$ as an example, its update works as follows:

$$\mathbf{\Delta}_2^{(t)} = \mathcal{P}_{\mathcal{S}}[\mathbf{\Delta}_2^{(t-1)} - \eta_t \nabla \mathcal{L}_{\text{GS}}(\mathbf{\Delta}_2^{(t-1)})] \tag{8}$$

where $\eta_t > 0$ is the learning rate for step $t$, and $\mathcal{P}_{\mathcal{S}}(\mathbf{a}) = \text{argmin}_{\mathbf{s} \in \mathcal{S}} \|\mathbf{s} - \mathbf{a}\|_2^2$ is the projection operator at $\mathbf{a}$ over the constraint set $\mathcal{S}$. The gradient $\nabla \mathcal{L}_{\text{GS}}(\mathbf{\Delta}_2^{(t-1)})$ can be calculated via chain rule, with a closed-form gradient over eigenvalues: for a real and symmetric matrix $\mathbf{L}$, the derivatives of its $k$-th eigenvalue $\lambda_k$ is $\partial \lambda_k / \partial \mathbf{L} = \mathbf{u}_k \mathbf{u}_k^\top$ (Rogers, 1970), where $\mathbf{u}_k$ is the corresponding eigenvector. Note that the derivative calculation requires distinct eigenvalues, which does not hold for graphs satisfying *automorphism* (Godsil, 1981). To avoid such cases, we add a small noise term to the adjacency matrix[1], e.g., $\mathbf{A} + \mathbf{C} \circ \mathbf{\Delta} + \varepsilon \times (\mathbf{N} + \mathbf{N}^\top)/2$, where each entry in $\mathbf{N}$ is sampled from a uniform distribution $\mathcal{U}(0, 1)$ and $\varepsilon$ is a very small constant. Such a noise addition will almost surely break the graph automorphism, thus enabling a valid gradient calculation of eigenvalues. The convergence of optimizing opposite $\mathbf{\Delta}_1$ and $\mathbf{\Delta}_2$ is empirically shown in Appendix D.7. Meanwhile, the behavior of spectral augmentation in the spatial domain is analyzed in Appendix D.6.

**Scalability.** For $T$ iterations, the time complexity of optimizing such a scheme is $\mathcal{O}(Tn^3)$ due to the eigen-decomposition $\text{eig}(\cdot)$ in $\mathcal{L}_{\text{GS}}$, which is prohibitively expensive for large graphs. To

---

[1] The form of $(\mathbf{N} + \mathbf{N}^\top)/2$ is to keep the perturbed adjacency matrix symmetric for undirected graphs.

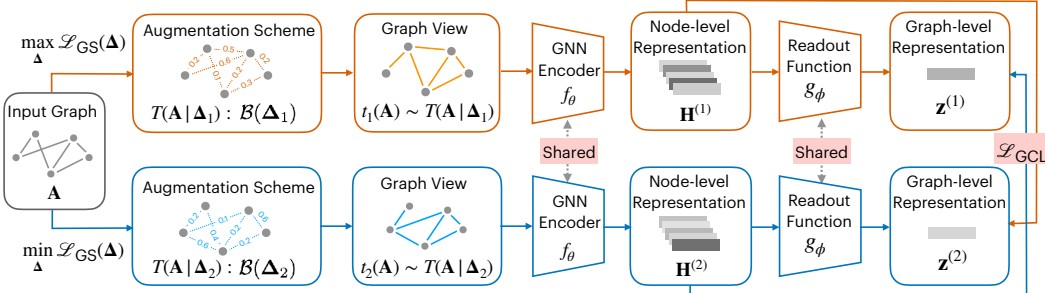

Figure 2: A GCL instantiation using SPAN. The augmentation scheme is pre-computed, and the augmented views are sampled from the scheme. The contrastive objective is to maximize the cross-level mutual information between node and graph representations.

reduce the computational cost, instead of conducting eigen-decomposition on the full graph spectrum, we appeal to selective eigen-decomposition on $K$ lowest- and highest-eigenvalues via the Lanczos Algorithm (Parlett & Scott, 1979). We consider both low and high eigenvalues in estimating the spectral change, because they play important roles in graph analysis and GNN design, as discussed in Appendix A. More detailed analysis of eigenvalues and the performance when varying $K$ on real-world graphs are provided in Appendix D.2. Th time complexity is reduced to $\mathcal{O}(TKn^2)$ [2], and the empirical running time for pre-computing the augmentation scheme is provided in Appendix D.1 We can further improve the scalability using practical treatments for large graphs (Qiu et al., 2020) (e.g., ego-nets sampling, batch training), which is left as our future work.

## 6 DEPLOYING SPAN IN GCL

We provide an instantiation of GCL with the proposed spectral augmentation, termed GCL-SPAN, as illustrated in Figure 2. Its detailed description can be found in Appendix B. Note that our focus is on the augmentation, and this GCL instantiation using widely-adopted designs is purposely to showcase how easily one can plug SPAN in most GCL frameworks.

The augmentation scheme can be first set up by pre-computing the probability matrices $\mathbf{\Delta}_1$ and $\mathbf{\Delta}_2$ based on Eq. 7. For each iteration of contrastive learning, we sample two augmented graphs for the input graph: $t_1(\mathbf{A}) \sim T(\mathbf{A}|\mathbf{\Delta}_1)$ and $t_2(\mathbf{A}) \sim T(\mathbf{A}|\mathbf{\Delta}_2)$. The augmented graphs are then fed into a GNN encoder $f_\theta$, which outputs two sets of node representations $\mathbf{H}^{(1)}, \mathbf{H}^{(2)} \in \mathbb{R}^{n \times d'}$. A readout pooling function $g_\phi$ is applied to aggregate and transform the node representations and obtain graph representations $\mathbf{z}^{(1)}, \mathbf{z}^{(2)} \in \mathbb{R}^{d'}$. Following (Velickovic et al., 2019; Hassani & Khasahmadi, 2020), given training graphs $\mathcal{G}$, we adopt the contrastive objective $\mathcal{L}_{\text{GCL}}$ that maximizes the cross-level correspondence between local node and global graph representations, which is to preserve local similarities and global invariants:

$$\text{GCL-TAGS} : \min_{\theta,\phi} \mathcal{L}_{\text{GCL}}(t_1(\mathbf{A}), t_2(\mathbf{A}), \theta, \phi) = -\frac{1}{|\mathcal{G}|} \sum_{G \in \mathcal{G}} \left( \frac{1}{n} \sum_{i=1}^{n} \left( I(\mathbf{H}_i^{(1)}, \mathbf{z}^{(2)}) + I(\mathbf{H}_i^{(2)}, \mathbf{z}^{(1)}) \right) \right)$$

$$\text{s.t. } t_i(\mathbf{A}) \sim T(\mathbf{A}|\mathbf{\Delta}_i), i \in \{1,2\}, \mathbf{\Delta}_1 = \text{argmax}_{\mathbf{\Delta} \in \mathcal{S}} \mathcal{L}_{\text{GS}}(\mathbf{\Delta}), \mathbf{\Delta}_2 = \text{argmin}_{\mathbf{\Delta} \in \mathcal{S}} \mathcal{L}_{\text{GS}}(\mathbf{\Delta}) \quad (9)$$

where $I(X_1, X_2)$ calculates the mutual information between variables $X_1$ and $X_2$, and it can be lower bounded by InfoNCE (Van den Oord et al., 2018; Poole et al., 2019). Specifically, denoting cosine similarity as $\cos(\cdot, \cdot)$, we estimate the mutual information as follows:

$$I(\mathbf{H}_i^{(a)}, \mathbf{z}^{(b)}) = \log \frac{\exp(\cos(\mathbf{H}_i^{(a)}, \mathbf{z}^{(b)}))}{\sum_{j=1}^{n} \exp(\cos(\widetilde{\mathbf{H}}_j, \mathbf{z}^{(b)}))} \quad (10)$$

where $a$ and $b$ index the augmented views, and $\widetilde{\mathbf{H}}$ is the node representations for negative examples in the batch. Note that the optimization of augmentation scheme is a one-time pre-computation thus does not introduce any extra complexity to the contrastive learning process. The proposed augmentation can be paired with other advanced choices in the contrastive paradigm, such as dynamic dictionary and moving-average encoder (He et al., 2020; Qiu et al., 2020) to fit different application scenarios.

---

[2]Since we only require to precompute $\mathbf{\Delta}_1$ and $\mathbf{\Delta}_2$ once, the time complexity is totally acceptable.

Table 1: Node classification performance in *unsupervised* setting. The metric is *accuracy (%)*. The best and second best results are highlighted with **bold** and underline respectively.

| | Dataset | Cora | Citeseer | PubMed | Wiki-CS | Amazon-Computer | Amazon-Photo | Coauthor-CS |
|---|---|---|---|---|---|---|---|---|
| | Raw-X | 48.93±0.00 | 50.81±0.00 | 68.33±0.00 | 71.98±0.00 | 73.81±0.00 | 78.53±0.00 | 90.37±0.00 |
| | S-GCN | 81.34±0.35 | 70.42±0.45 | 79.82±0.41 | 77.19±0.12 | 86.51±0.54 | 92.42±0.16 | 93.03±0.31 |
| | R-GCN | 56.44±0.24 | 63.52±0.25 | 73.92±0.32 | 72.95±0.58 | 82.46±0.38 | 90.08±0.48 | 90.64±0.29 |
| Baselines | GRACE | 83.33±0.43 | 72.10±0.54 | 78.72±0.13 | 80.14±0.48 | 89.53±0.35 | 92.78±0.30 | 91.12±0.20 |
| | BGRL | 83.63±0.38 | 72.52±0.40 | 79.83±0.25 | 79.98±0.13 | **90.34±0.19** | 93.17±0.30 | 93.31±0.13 |
| | GBT | 80.24±0.42 | 69.39±0.56 | 78.29±0.43 | 76.65±0.62 | 88.14±0.33 | 92.63±0.44 | 92.95±0.17 |
| | MVGRL | 85.16±0.52 | 72.14±1.35 | 80.13±0.84 | 77.52±0.08 | 87.52±0.11 | 91.74±0.07 | 92.11±0.12 |
| | GCA | 83.67±0.44 | 71.48±0.26 | 78.87±0.49 | 78.35±0.05 | 88.94±0.15 | 92.53±0.16 | 93.10±0.01 |
| | GMI | 83.02±0.33 | 72.45±0.12 | 79.94±0.25 | 74.85±0.08 | 82.21±0.31 | 90.68±0.17 | 91.08±0.56 |
| | DGI | 82.34±0.64 | 71.85±0.74 | 76.82±0.61 | 75.35±0.14 | 83.95±0.47 | 91.61±0.22 | 92.15±0.63 |
| | GCL-SPAN | **85.86±0.57** | **72.76±0.63** | **81.54±0.24** | **82.13±0.15** | 90.09±0.32 | **93.52±0.26** | **93.91±0.24** |

## 7 EXPERIMENTS

**Experiment Setup:** We perform evaluations for node classification, graph classification and regression tasks under unsupervised learning, transfer learning and adversarial attack settings. Following prior works (Zhu et al., 2021; Thakoor et al., 2021; Veličković et al., 2019) , we use GCN (for node prediction tasks) and GIN (for graph prediction tasks) as the base encoder $f_\theta$ for all methods to demonstrate the performance gain from contrastive learning. We adopt the following evaluation protocol for downstream tasks (Suresh et al., 2021): based on the representations given by the encoder, we train and evaluate a Logistic classifier or a Ridge regressor. We repeat our experiments for 10 times and report the mean and standard derivation of the evaluation metrics. The experimental details about datasets, baselines and configurations can be found in Appendix C.

### 7.1 UNSUPERVISED LEARNING

To evaluate the quality of learned representations, a linear model is trained for the downstream tasks using these learned representations as features and the resulting prediction performance is reported. The effectiveness of SPAN is evaluated on both node- and graph-level prediction tasks.

**Node Classification Task.** We first evaluate the effectiveness of SPAN on node classification datasets ranging from citation networks to co-purchase networks. We compare against GCL methods that augment topology with uniformly random edge perturbation (e.g., GRACE (Zhu et al., 2020), BGRL (Thakoor et al., 2021), GBT (Bielak et al., 2021)), centrality (GCA (Zhu et al., 2021)), diffusion matrix (MVGRL (Hassani & Khasahmadi, 2020)) and the original graph (e.g., GMI (Peng et al., 2020) and DGI (Veličković et al., 2019)). As a reference, we also consider a fully semi-supervised GCN (*S-GCN*), a randomly initialized untrained GCN (*R-GCN*) and using the raw node features as node representations (*Raw-X*). All the methods exploit a 2-layer GCN as backbone and a downstream linear classifier with the same hyper-parameters for a fair comparison. We adopt random feature masking in SPAN, following the setup in SOTA works (Zhu et al., 2021; Bielak et al., 2021).

Table 1 compares the instantiation GCL-SPAN with baselines. Comparing GCL-SPAN with MVGRL and GCA which augment graphs via domain knowledge (e.g., node centrality or graph diffusion), the performance gain suggests the advantage of the spectral augmentation over previous domain-knowledge based heuristics. Meanwhile, SPAN is shown to be more effective than the uniformly random augmentation adopted in GRACE, BGRL and GBT. To demonstrate the direct gain of SPAN, we also compare the performance of directly plugging our proposed augmentation into several existing frameworks in Appendix D.3; and to compare the effectiveness of different augmentation schemes, we compare the performance when using different augmentation schemes for our GCL instantiation in Appendix D.5. These experiments show promising improvement by our proposed SPAN.

**Graph Prediction Task.** We test on multiple graph classification and regression datasets ranging from social networks, chemical molecules to biological networks. We compare SPAN with five GCL methods including InfoGraph (Sun et al., 2019), GraphCL (You et al., 2020a), MVGRL (Hassani & Khasahmadi, 2020), AD-GCL (with fixed regularization weight) (Suresh et al., 2021) and JOAO (v2) (You et al., 2021). We use a 5-layer GIN encoder for all methods, including a semi-supervised *S-GIN* and a randomly initialized *R-GIN*. A readout function with a graph pooling layer and a 2-layer

Table 2: Graph representation learning performance in *unsupervised* setting. TOP shows the biochemical and social network classification results on TU datasets (measured by *accuracy%*). BOTTOM shows the regression (measured by *RMSE*) and classification (measured by *ROC-AUC%*) results on OGB datasets. The best and second best results are highlighted with **bold** and underline respectively.

| Dataset | Biochemical Molecules | | | | Social Networks | | | | |
|---|---|---|---|---|---|---|---|---|---|
| | NCI1 | PROTEINS | MUTAG | DD | COLLAB | RDT-B | RDT-M5K | IMDB-B | IMDB-M |
| S-GIN | 78.27±1.35 | 72.39±2.76 | 90.41±4.61 | 74.87±3.56 | 74.82±0.92 | 86.79±2.04 | 53.28±3.17 | 71.83±1.93 | 48.46±2.31 |
| R-GIN | 62.98±0.10 | 69.03±0.33 | 87.61±0.39 | 74.22±0.30 | 63.08±0.10 | 58.97±0.13 | 27.52±0.61 | 51.86±0.33 | 32.81±0.57 |
| InfoGraph | 68.13±0.59 | 72.57±0.65 | 87.71±1.77 | 75.23±0.39 | 70.35±0.64 | 78.79±2.14 | 51.11±0.55 | 71.11±0.88 | 48.66±0.67 |
| GraphCL | 68.54±0.55 | 72.86±1.01 | 88.29±1.31 | 74.70±0.70 | 71.26±0.55 | 82.63±0.99 | 53.05±0.40 | 70.80±0.77 | 48.49±0.63 |
| MVGRL | 68.68±0.42 | 74.02±0.32 | 89.24±1.31 | 75.20±0.55 | 73.10±0.56 | 81.20±0.69 | 51.87±0.65 | 71.84±0.78 | 50.84±0.92 |
| AD-GCL | 69.67±0.51 | 73.59±0.65 | **89.25±1.45** | 74.49±0.52 | 73.32±0.61 | **85.52±0.79** | 53.00±0.82 | 71.57±1.01 | 49.04±0.53 |
| JOAO | **72.99±0.75** | 71.25±0.85 | 85.20±1.64 | 66.91±1.75 | 70.40±2.21 | 78.35±1.38 | 45.57±2.86 | 71.60±0.86 | 51.14±0.69 |
| GCL-SPAN | 71.43±0.49 | **75.78±0.41** | 89.12±0.76 | **75.78±0.52** | **75.01±0.45** | 83.62±0.64 | **54.10±0.49** | **73.65±0.69** | **52.16±0.72** |

| Dataset | Regression (Metric: RMSE) | | | Classification (Metric: ROC-AUC%) | | | | |
|---|---|---|---|---|---|---|---|---|
| | molesol | mollipo | molfreesolv | molbace | molbbbp | molclintox | moltox21 | molsider |
| S-GIN | 1.173±0.057 | 0.757±0.018 | 2.755±0.349 | 72.97±4.00 | 68.17±1.48 | 88.14±2.51 | 74.91±0.51 | 57.60±1.40 |
| R-GIN | 1.706±0.180 | 1.075±0.022 | 7.526±2.119 | 75.07±2.23 | 64.48±2.46 | 72.29±4.15 | 71.53±0.74 | 62.29±1.12 |
| InfoGraph | 1.344±0.178 | 1.005±0.023 | 10.005±4.819 | 74.74±3.64 | 66.33±2.79 | 64.50±5.32 | 69.74±0.57 | 60.54±0.90 |
| GraphCL | 1.272±0.089 | 0.910±0.016 | 7.679±2.748 | 74.32±2.70 | 68.22±1.89 | 74.92±4.42 | 72.40±1.01 | 61.76±1.11 |
| MVGRL | 1.433±0.145 | 0.962±0.036 | 9.024±1.982 | 74.20±2.31 | 67.24±1.39 | 73.84±4.25 | 70.48±0.83 | 61.94±0.94 |
| AD-GCL | **1.217±0.087** | 0.842±0.028 | 5.150±0.624 | 76.37±2.03 | 68.24±1.47 | **80.77±3.92** | 71.42±0.73 | 63.19±0.95 |
| JOAO | 1.285±0.121 | 0.865±0.032 | 5.131±0.722 | 74.43±1.94 | 67.62±1.29 | 78.21±4.12 | 71.83±0.92 | 62.73±0.92 |
| GCL-SPAN | 1.218±0.052 | **0.802±0.019** | 4.531±0.463 | 76.74±2.02 | 69.59±1.34 | 80.28±2.42 | 72.83±0.62 | 64.87±0.88 |

MLP is applied to generate graph representations. We adopt the given data split for OGB dataset, and use 10-fold cross validation for TU dataset as it does not provide such a split.

Table 2 summarizes the graph prediction performance. GCL-SPAN gives the best results on 13 out of 17 datasets, of which 10 are significantly better than others. Compared with GraphCL and JOAO which select the best combination of augmentations for each dataset from a pool of methods including edge perturbation, node dropping and subgraph sampling, GCL-SPAN using only edge perturbation based augmentation still outperforms them. This suggests the effectiveness of graph spectrum in guiding topology augmentation. Compared with MVGRL, our performance gain mainly comes from the augmentation scheme, as these two methods share similar contrastive objectives, and our augmentation guided by graph spectrum is clearly more effective than the widely adopted uniformly random augmentation. While AD-GCL and GCL-SPAN follow a similar principle to remove edges that carry non-important and redundant information, GCL-SPAN is more flexible since the augmentation scheme is optimized in an independent pre-computation step without interfering with the contrastive learning procedure.

## 7.2 TRANSFER LEARNING

This experiment evaluates the generalizability of the GNN encoder, which is pre-trained on a source dataset and re-purposed on a target dataset. Table 3 reports the performance on chemical and

Table 3: Graph classification performance in *transfer learning* setting on molecular classification task. The metric is *ROC-AUC%*. The best and second best results are shown in **bold** and underline.

| Dataset | Pre-Train | ZINC-2M | | | | | | | | PPI-306K |
|---|---|---|---|---|---|---|---|---|---|---|
| | Fine-Tune | BBBP | Tox21 | SIDER | ClinTox | BACE | HIV | MUV | ToxCast | PPI |
| No-Pre-Train-GIN | | 65.8±4.5 | 74.0±0.8 | 57.3±1.6 | 58.0±4.4 | 70.1±5.4 | 75.3±1.9 | 71.8±2.5 | 63.4±0.6 | 64.8±1.0 |
| InfoGraph | | 68.8±0.8 | 75.3±0.5 | 58.4±0.8 | 69.9±3.0 | 75.9±1.6 | 76.0±0.7 | **75.3±2.5** | 62.7±0.4 | 64.1±1.5 |
| GraphCL | | 69.7±0.7 | 73.9±0.7 | 60.5±0.9 | 76.0±2.7 | 75.4±1.4 | **78.5±1.2** | 69.8±2.7 | 62.4±0.6 | 67.9±0.9 |
| MVGRL | | 69.0±0.5 | 74.5±0.6 | 62.2±0.6 | 77.8±2.2 | 77.2±1.0 | 77.1±0.6 | 73.3±1.4 | 62.6±0.5 | 68.7±0.7 |
| AD-GCL | | 70.0±1.1 | 76.5±0.8 | 63.3±0.8 | 79.8±3.5 | 78.5±0.8 | 78.3±1.0 | 72.3±1.6 | 63.1±0.7 | 68.8±1.3 |
| JOAO | | **71.4±0.9** | 74.3±0.6 | 60.5±0.7 | **81.0±1.6** | 75.5±1.3 | 77.5±1.2 | 73.7±1.0 | 63.2±0.5 | 64.0±1.6 |
| GCL-SPAN | | 70.0±0.7 | **78.0±0.5** | **64.7±0.5** | 80.7±2.1 | **79.9±0.7** | 77.8±0.6 | 73.8±0.9 | **64.2±0.4** | **70.0±0.8** |

Table 4: Node classification performance on Cora in *adversarial attack* setting (measured by *accuracy%*). The best and second best results are highlighted with **bold** and underline respectively.

| Attack | Clean | Random | | DICE | | GF-Attack | | Mettack | |
|---|---|---|---|---|---|---|---|---|---|
| Ratio $\sigma$ | | 0.05 | 0.2 | 0.05 | 0.2 | 0.05 | 0.2 | 0.05 | 0.2 |
| S-GCN | 81.34±0.35 | 81.11±0.32 | 80.02±0.36 | 79.42±0.37 | 78.37±0.42 | 80.12±0.33 | 79.43±0.32 | 50.29±0.41 | 31.04±0.48 |
| GRACE | 83.33±0.43 | 83.23±0.38 | 82.57±0.48 | 81.28±0.39 | 80.72±0.44 | 82.59±0.35 | 80.23±0.38 | 67.42±0.59 | 55.26±0.53 |
| BGRL | 83.63±0.38 | 83.12±0.34 | 83.02±0.39 | 82.83±0.48 | 81.92±0.39 | 82.10±0.37 | 80.98±0.42 | 70.23±0.48 | 60.42±0.54 |
| GBT | 80.24±0.42 | 80.53±0.39 | 80.20±0.35 | 80.32±0.32 | 80.20±0.34 | 79.89±0.41 | 78.25±0.49 | 63.26±0.69 | 53.89±0.55 |
| MVGRL | 85.16±0.52 | 85.28±0.49 | 84.21±0.42 | 83.78±0.35 | 83.02±0.40 | 83.79±0.39 | 82.46±0.52 | 73.43±0.53 | 61.49±0.56 |
| GCA | 83.67±0.44 | 83.33±0.46 | 82.49±0.37 | 82.20±0.32 | 81.82±0.45 | 81.83±0.36 | 79.89±0.47 | 58.25±0.68 | 49.25±0.62 |
| GMI | 83.02±0.33 | 83.14±0.38 | 82.12±0.44 | 82.42±0.44 | 81.13±0.49 | 82.13±0.39 | 80.26±0.48 | 60.59±0.54 | 53.67±0.68 |
| DGI | 82.34±0.64 | 82.10±0.58 | 81.03±0.52 | 80.48±0.38 | 79.89±0.43 | 81.30±0.54 | 79.88±0.58 | 71.42±0.63 | 63.93±0.58 |
| GCL-SPAN | **85.86±0.57** | **86.29±0.52** | **86.21±0.78** | **85.52±0.59** | **84.30±0.63** | **85.08±0.77** | **84.28±0.82** | **77.28±0.82** | **69.92±0.83** |

(The GRACE–DGI rows are grouped under the label "Baselines".)

biological graph classification datasets from (Hu et al., 2020b). Appendix C.5 studies an even more challenging setting (Qiu et al., 2020) where the encoder is pre-trained on social networks and transferred to multiple out-of-domain tasks. A reference model without pre-training (*No-Pre-Train-GIN*) is compared to demonstrate the gain of pre-training. GCL-SPAN is shown to be more effective in learning generalizable encoders, which supports our augmentation principle: by perturbing edges that cause large spectral changes, the encoder is pre-trained to ignore unreliable structural information, such that the relationship between such information and downstream labels can be removed to mitigate the overfitting issue. The generalizability of the GNN encoder on molecule classification depends on the structural fingerprints such as *subgraphs* (Duvenaud et al., 2015). JOAO and GraphCL using *subgraph* sampling augmentation is outperformed by GCL-SPAN, which suggests that the graph spectrum could be another important fingerprint to study chemical and biological molecules.

## 7.3 ADVERSARIAL ROBUSTNESS

This setting demonstrates the robustness of GCL when the input graph is adversarially poisoned. The augmentation scheme is optimized and the representations are learned from graphs poisoned by different structural attack strategies, including *Random* (which randomly flips edges), *DICE* (Waniek et al., 2018) (which deletes edges internally and connects nodes externally across classes), *GF-Attack* (Chang et al., 2020) (which maximizes a low-rank matrix approximation loss) and *Mettack* (Zügner & Günnemann, 2019) (which maximizes the training loss via meta-gradients). We test the perturbation ratios $\sigma \in \{0.05, 0.2\}$: $\sigma \times m$ edges are flipped for a graph with $m$ edges.

Table 4 reports the node classification performance under the adversarial attacks. The encoders learned by GCL methods with graph augmentations are generally more robust to perturbed graphs compared with S-GCN. GCL-SPAN outperforms baselines with a clear margin, which demonstrates a good property of SPAN: even though it is not explicitly designed for adversarial robustness, the encoder can stay invariant to the adversarially perturbed graph if its spectrum falls into the range captured by the opposite-direction augmentation scheme. Compared with the parallel efforts in designing robust GNNs (Entezari et al., 2020; Zhu et al., 2019; Jin et al., 2020b), we provide a new insight using graph spectrum as a tool to study graph invariance under perturbations.

## 8 CONCLUSION

In this work, we proposed a principled spectral augmentation scheme which generates topology augmentations by perturbing graph spectrum. Our principle is that a well-behaving GNN encoder should preserve *spectral invariance* to sensitive structures that could cause large changes on graph spectrum. To achieve this goal, we search for the augmentation scheme that would mostly change the graph spectrum of the input graph, which further leads to an opposite-direction augmentation scheme that changes the graph spectrum towards opposite directions in two views. The proposed augmentation can be paired with various GCL frameworks, and the extensive experiments demonstrate the performance gain by the proposed augmentation method. Currently we only focus on topology augmentation, and ignore its interplay with node features, which is another important dimension in GCL. Recent studies show the relationship between graph homophily/heterophily and GNNs, which suggests a future effort to explore the alignment between graph topology and node features in GCL.

## 9 ACKNOWLEDGEMENTS

This work is supported by the National Science Foundation under grant IIS-2007492, IIS-1838615 and IIS-1718216.

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

# A A REVIEW OF GRAPH SPECTRUM

Graph spectrum plays a significant role in analyzing graph property (e.g., connectivity, cluster structure, diameter etc.) and is the foundation of spectral filters in GNNs. This motivates us to guide our proposed topology augmentation method using graph spectrum.

**Graph Spectrum and Graph Property.** The graph spectrum summarizes important properties related to a graph's global structure, which has been studied in graph spectral theory. We list some widely discussed properties revealed by graph spectrum to support our design: graph spectrum can be used as a comprehensive proxy for capturing graph properties in GCL.

- *Algebraic connectivity* (Chung & Graham, 1997), also known as Fiedler eigenvalue, of a graph is the second-smallest eigenvalue (counting multiple eigenvalues separately) of the Laplacian matrix. This eigenvalue is greater than 0 if and only if the graph is a connected graph. A corollary is that the number of times 0 appears as an eigenvalue in the Laplacian is the number of connected components in the graph. The magnitude of this value reflects how well connected the overall graph is.

- *Diameter* of a graph can be upper and lower bounded from its spectrum (Chung & Graham, 1997). If the graph has $r$ distinct eigenvalues, its diameter $d$ is at most $r - 1$. Meanwhile, if the graph has $m$ edges and $n$ nodes, we can bound the diameter by the first and second smallest non-zero eigenvalues as $1/2m\lambda_1 \geq d \geq 4/n\lambda_2$. For all $k \geq 2$, we also have $d \leq k \log n/\lambda_k$.

- *Clusterability* of a graph reveals how easy it is to partition the graph, which can be captured by the differences between the smallest successive eigenvalues of connected graphs. The difference between the first two eigenvalues, often referred to as the spectral gap, upper and lower bounds the graph expansion and conductance by the Cheeger inequality (Kahale, 1995). Nevertheless, analogous results also hold for higher-order eigenvalues (Lee et al., 2014).

- *Diffusion distance* (Hammond et al., 2013) between two nodes $v_i$ and $v_j$ can be defined as $\mathcal{D}(v_i, v_j) = \|[\phi(\mathbf{L})]_{i,:} - [\phi(\mathbf{L})]_{j,:}\|_2^2 = \sum_{l=1}^n \phi(\lambda_l)^2(\mathbf{u}_l[i] - \mathbf{u}_l[j])^2$, where $\phi(\mathbf{L}) = \mathbf{U}\phi(\mathbf{\Lambda})\mathbf{U}^\top$ and $\phi(x)$ is a decreasing kernel function such as $\phi(x) = e^{-tx}$. Therefore, a map that separates nodes with a specific diffusion distance is obtained when embedding graph nodes using eigenvectors.

**Graph Spectrum and GNNs.** By viewing GNN models from a signal processing perspective, the normalized Laplacian $\mathbf{L}_{\text{norm}}$ serves as a graph shift operator and defines the frequency domain of a graph. As a result, its eigenvectors $\mathbf{U}$ can be considered as the graph Fourier bases, and the eigenvalues $\mathbf{\Lambda}$ (a.k.a., graph spectrum) correspond to the frequency components. Take one column of node features $\mathbf{X}$ as an example of graph signal, which can be compactly represented as $\mathbf{x} \in \mathbb{R}^n$. The graph Fourier transform of graph signal $\mathbf{x}$ is given by $\hat{\mathbf{x}} = \mathbf{U}^\top \mathbf{x}$ and the inverse graph Fourier transform is then $\mathbf{x} = \mathbf{U}\hat{\mathbf{x}}$. The graph signals in the Fourier domain are filtered by amplifying or attenuating the frequency components $\mathbf{\Lambda}$.

At the essence of GNNs is the *spectral convolution*, which can be defined as the multiplication of a signal vector $\mathbf{x}$ with a spectral filter $g_\phi$ in the Fourier domain (Defferrard et al., 2016):

$$g_\phi(\mathbf{L}) \star \mathbf{x} = g(\mathbf{U}\mathbf{\Lambda}\mathbf{U}^\top)\mathbf{x} = \mathbf{U}g_\phi(\mathbf{\Lambda})\mathbf{U}^\top\mathbf{x} \tag{11}$$

The filter $g_\phi$ defines a smooth transformation function of the graph spectrum. One can apply a spectral filter and graph Fourier transformation to manipulate graph signals in various way, such as smoothing and denoising (Schaub & Segarra, 2018), abnormally detection (Miller et al., 2011) and clustering (Wai et al., 2018). Here we show how the spectral convlution is defined in two popular GNNS used in our experiments: GCN (Kipf & Welling, 2017) and GIN (Xu et al., 2019).

The vanilla GCN (Kipf & Welling, 2017) is a first-order approximation of Chebyshev polynomial filter (Hammond et al., 2011) with $g_\phi(\mathbf{\Lambda}) = (2 - \mathbf{\Lambda})\phi$, and the corresponding convolution for $d$-dimensional signal $\mathbf{X}$ is:

$$g_\phi^{\text{GCN}}(\mathbf{L}) \star \mathbf{X} = \mathbf{U}(2 - \mathbf{\Lambda})\mathbf{U}^\top\mathbf{X}\mathbf{\Phi} = (\mathbf{I}_n + \mathbf{D}^{-1/2}\mathbf{A}\mathbf{D}^{-1/2})\mathbf{X}\mathbf{\Phi} = \tilde{\mathbf{D}}^{-1/2}\tilde{\mathbf{A}}\tilde{\mathbf{D}}^{-1/2}\mathbf{X}\mathbf{\Phi} \tag{12}$$

where $\mathbf{\Phi} \in \mathbf{R}^{d \times d'}$ is the matrix of spectral filter parameters, and a renormalization trick $\mathbf{I}_n + \mathbf{D}^{-1/2}\mathbf{A}\mathbf{D}^{-1/2} \rightarrow \tilde{\mathbf{D}}^{-1/2}\tilde{\mathbf{A}}\tilde{\mathbf{D}}^{-1/2}$ is applied by adding self-loop $\tilde{\mathbf{A}} = \mathbf{A} + \mathbf{I}_n$. GIN (Xu et al.,

2019) with equal discriminative power as WL test designs spectral convolution as:

$$g_\phi^{\text{GIN}}(\mathbf{L}) \star \mathbf{X} = \mathbf{U}(2 + \epsilon - \mathbf{\Lambda})\mathbf{U}^\top \mathbf{X}\mathbf{\Phi} = (\mathbf{I}_n(1 + \epsilon) + \mathbf{D}^{-1/2}\mathbf{A}\mathbf{D}^{-1/2})\mathbf{X}\mathbf{\Phi} \qquad (13)$$

where $\epsilon$ can be a learnable parameter or a fixed scalar. Since the spectral filters $g_\phi(\mathbf{\Lambda})$ are the key in graph convolutions to encode graph signals that are transformed in the Fourier domain. The output of the spectral filters is then transformed back to the spatial domain to generate node representations. Therefore, we aim to augment graphs to influence the graph spectrum and the filtered graph signals, such that the encoder with altered spectral filters is encouraged to stay invariant to such perturbations through GCL.

Some recent literature has shown that graph spectrum is closely related to GNNs' performance. For example, it is proved that the generalization gap of a single layer GCN model trained via $T$-step SGD is $\mathcal{O}(\lambda_n^{2T})$, where $\lambda_n$ is the largest eigenvalue of graph Laplacian (Verma & Zhang, 2019). Meanwhile, Weinberger et al. (2006) proved that a generalization estimate is inversely proportional to the second smallest eigenvalue of the graph Laplacian $\lambda_2$. More recent works (Chang et al., 2021b; Luan et al., 2021; Bo et al., 2021) reveal that both low and high eigenvalues are important for better graph representation learning to capture both smoothly varying signals and diversified information.

## B    ALGORITHM OF DEPLOYING SPAN IN GCL

Algorithm 1 illustrates the detailed steps of deploying SPAN in an instantiation of GCL. Note that only one training graph is included (e.g., $L = 1$) for node representations learning, and multiple graphs are used (e.g., $L \geq 2$) for graph representation learning.

## C    EXPERIMENT SETUP DETAILS

This section includes the detailed setup for all experiments, including the procedure of conducting pre-analysis, datasets, baselines, and hyper-parameter settings. The experiments were performed on Nvidia GeForce RTX 2080Ti (12GB) GPU cards for most datasets, and RTX A6000 (48GB) cards for PubMed and Coauthor-CS datasets. Optimizing memory use for large graphs will be our future work.

### C.1    PRE-ANALYSIS EXPERIMENT OF FIGURE 1

We now introduce the detailed information to reproduce Figure 1 on Cora. This experiment is to show that uniformly random edge perturbation adopted in many GCL methods is not effective enough to capture structural invariant regarding essential graph properties. In contrast to the uniform edge perturbation, we created a cluster based strategy as follows: We first grouped the edges among nodes by whether the end nodes belong to the same cluster (which is given by the spectral clustering algorithm). For inter-cluster edges, we assign a larger removing probability, while for intra-cluster edges we assign a smaller removing probability. Note that in expectation, we remove the same amount of edges as the uniformly *random* strategy, but allocate different probabilities to these two groups of edges.

Specifically, an edge removing ratio $\sigma$ indicated by the x-axis of Figure 1 represents the augmentation strength: for an input graph with $m$ edges, we remove $\sigma \cdot m$ edges to generate an augmented graph. For the uniformly random augmentation method (*Uniform*), each edge is assigned an equal removing probability as $\sigma$; for the cluster-based augmentation heuristic (*Clustered*), given $m_{\text{inter}}$ inter-cluster edges and $m_{\text{intra}} = m - m_{\text{inter}}$ intra-cluster edges, we increase the removing probability of each inter-cluster edge as $\sigma_{\text{inter}} = \min\{1.2\sigma, \sigma \cdot m/m_{\text{inter}}\}$, and the removing probability of each intra-cluster edge is decreased to $\sigma_{\text{intra}} = (\sigma \cdot m - \sigma_{\text{inter}} \cdot m_{\text{inter}})/m_{\text{intra}}$ to make sure that in expectation $\sigma \cdot m$ edges are removed as in the uniform strategy.

When conducting the contrastive learning procedure, one augmentation branch used the original graph, and the other branch adopted either the uniform or the clustered strategy with a random feature masking with ratio 0.3. For these two GCL methods based on different augmentation strategies, the experiment setup is as follows: both methods used a GCN encoder with the same architecture and hyper-parameters (e.g., 2 convolutional layers with the embedding dimension of 32). Both performed 1000 training iterations to obtain node representations, whose quality was evaluated by using them as features for a downstream linear Logistic classifier.

---

**Algorithm 1:** Deploying SPAN in an instantiation of GCL

---

**Input** : Data $\{G_l = (\mathbf{X}_l, \mathbf{A}_l) \sim \mathcal{G} | l = 1 \cdots, L\}$; GNN encoder $f_\theta$; Readout function $g_\phi$.
**Params** : Augmentation optimization step $M$ and learning rate $\eta$;
         Contrastive learning step $N$ and learning rate $\beta$.
**Output** : Trained encoder $f_{\theta^*}$ and readout function $f_{\phi^*}$.

---

1   /* Lower-level optimization for augmentation scheme in Eq. 9 */
2   **for** *each graph* $l \leftarrow 1$ **to** $L$ **do**
3     Initialize Bernoulli parameters for each graph: $\boldsymbol{\Delta}_{l,1}^{(0)} \in [0,1]^{n \times n}, \boldsymbol{\Delta}_{l,2}^{(0)} \in [0,1]^{n \times n}$;
4     **for** $t \leftarrow 1$ **to** $M$ **do**
5       /* Optimize one direction of scheme: $\max_{\boldsymbol{\Delta} \in \mathcal{S}} \mathcal{L}_{\text{GS}}(\boldsymbol{\Delta})$ */
6       $\mathcal{L}_{\text{GS}}(\boldsymbol{\Delta}_{l,1}^{(t-1)}) = \|\text{eig}(\text{Lap}(\mathbf{A} + \mathbf{C} \circ \boldsymbol{\Delta}_{l,1}^{(t-1)}))\|_2^2$;
7       $\boldsymbol{\Delta}_{l,1}^{(t)} \leftarrow \mathcal{P}_{\mathcal{S}}[\boldsymbol{\Delta}_{l,1}^{(t-1)} + \eta \nabla \mathcal{L}_{\text{GS}}(\boldsymbol{\Delta}_{l,1}^{(t-1)})]$;
8       /* Optimize the other direction of scheme: $\min_{\boldsymbol{\Delta} \in \mathcal{S}} \mathcal{L}_{\text{GS}}(\boldsymbol{\Delta})$ */
9       $\mathcal{L}_{\text{GS}}(\boldsymbol{\Delta}_{l,2}^{(t-1)}) = \|\text{eig}(\text{Lap}(\mathbf{A} + \mathbf{C} \circ \boldsymbol{\Delta}_{l,2}^{(t-1)}))\|_2^2$;
10      $\boldsymbol{\Delta}_{l,2}^{(t)} \leftarrow \mathcal{P}_{\mathcal{S}}[\boldsymbol{\Delta}_{l,2}^{(t-1)} - \eta \nabla \mathcal{L}_{\text{GS}}(\boldsymbol{\Delta}_{l,2}^{(t-1)})]$ based on Eq. 8;
11     **end**
12     $\boldsymbol{\Delta}_{l,1} \leftarrow \boldsymbol{\Delta}_{l,1}^{(M)}, \boldsymbol{\Delta}_{l,2} \leftarrow \boldsymbol{\Delta}_{l,2}^{(M)}$;
13 **end**

14 /* Upper-level optimization for contrastive learning in Eq. 9 */
15 Initialize encoder and readout function: $\theta^{(0)}, \phi^{(0)}$;
16 **for** $t \leftarrow 1$ **to** $N$ **do**
17     Sample a batch of graphs $\{G_1, \cdots, G_Q\}$;
18     /* Sample augmented views for this graph based on Eq. 4 */
19     **for** $l \leftarrow 1$ **to** $Q$ **do**
20       Sample perturbations from Bernoulli distributions: $\mathbf{E}_{l,1} \sim \mathcal{B}(\boldsymbol{\Delta}_{l,1}), \mathbf{E}_{l,2} \sim \mathcal{B}(\boldsymbol{\Delta}_{l,2})$;
21       Calculate topology augmentations: $\mathbf{A}_{l,1} = \mathbf{A} + \mathbf{C} \circ \mathbf{E}_{l,1}, \mathbf{A}_{l,2} = \mathbf{A} + \mathbf{C} \circ \mathbf{E}_{l,2}$;
22       Randomly mask node features to obtain $\mathbf{X}_{l,1}, \mathbf{X}_{l,2}$ following (Zhu et al., 2021; Bielak et al., 2021) if applicable;
23       Two graph views are generated as $G_{l,1} = (\mathbf{X}_{l,1}, \mathbf{A}_{l,1}), G_{l,2} = (\mathbf{X}_{l,2}, \mathbf{A}_{l,2})$
24     **end**
25     /* Optimize contrastive objective $\min_{\theta,\phi} \mathcal{L}_{\text{GCL}}$ */
26     Define $\mathcal{L}(G_{l,1}, G_{l,2}, \theta, \phi) = \frac{1}{n} \sum_{i=1}^{n} \left( I(\mathbf{H}_i^{(1)}, \mathbf{z}^{(2)}) + I(\mathbf{H}_i^{(2)}, \mathbf{z}^{(1)}) \right)$ for $G_l$;
27     Calculate objective: $\mathcal{L}_{\text{GCL}}(\theta^{(t-1)}, \phi^{(t-1)}) = -\frac{1}{Q} \sum_{l=1}^{Q} \mathcal{L}(G_{l,1}, G_{l,2}, \theta^{(t-1)}, \phi^{(t-1)})$;
28     Update the encoder: $\theta^{(t)} \leftarrow \theta^{(t-1)} - \beta \nabla_\theta \mathcal{L}_{\text{GCL}}(\theta^{(t-1)}, \phi^{(t-1)})$;
29     Update the readout function: $\phi^{(t)} \leftarrow \phi^{(t-1)} - \beta \nabla_\phi \mathcal{L}_{\text{GCL}}(\theta^{(t-1)}, \phi^{(t-1)})$;
30 **end**
    **Output** : Encoder $f_{\theta^{(N)}}$ and readout function $h_{\phi^{(N)}}$

---

Table 5: Node classification dataset. The metric is *accuracy*.

| Data Name | #Nodes | #Edges | #Features | #Classes | Cluster Coefficient |
|---|---|---|---|---|---|
| Cora | 2,708 | 5,278 | 1,433 | 7 | 0.2407 |
| Citeseer | 3,327 | 4,552 | 3,703 | 6 | 0.1415 |
| PubMed | 19,717 | 44,324 | 500 | 3 | 0.0602 |
| Wiki-CS | 11,701 | 215,863 | 300 | 10 | 0.4527 |
| Amazon-Computers | 13,752 | 245,861 | 767 | 10 | 0.3441 |
| Amazon-Photos | 7,650 | 119,081 | 745 | 8 | 0.4040 |
| Coauthor-CS | 18,333 | 81,894 | 6,805 | 15 | 0.3425 |

Table 6: TU Benchmark Datasets (Morris et al., 2020) for graph classifcation task in unsupervised learning setting. The metric used for classification task is *accuracy*.

| Data Type | Name | #Graphs | Avg. #Nodes | Avg. #Edges | #Classes |
|---|---|---|---|---|---|
| Biochemical Molecules | NCI1 | 4,110 | 29.87 | 32.30 | 2 |
| | PROTEINS | 1,113 | 39.06 | 72.82 | 2 |
| | MUTAG | 188 | 17.93 | 19.79 | 2 |
| | DD | 1,178 | 284.32 | 715.66 | 2 |
| Social Networks | COLLAB | 5,000 | 74.5 | 2457.78 | 3 |
| | REDDIT-BINARY | 2,000 | 429.6 | 497.75 | 2 |
| | REDDIT-MULTI-5K | 4,999 | 508.8 | 594.87 | 5 |
| | IMDB-BINARY | 1,000 | 19.8 | 96.53 | 2 |
| | IMDB-MULTI | 1,500 | 13.0 | 65.94 | 3 |

Figure 1a reports the mean and standard derivation of *F1 score* for 10 runs with different random seeds, which measures the downstream task performance. Meanwhile, we calculated the eigenvalues of the normalized Laplacian matrix of the input graph ($\Lambda$), the augmented graphs with uniform strategy ($\Lambda'_{\text{uniform}}$) and the augmented graphs with clustered strategy ($\Lambda'_{\text{clustered}}$). Figure 1b reports the $L_2$ distance of eigenvalues between the input and augmented graphs (e.g., $\|\Lambda - \Lambda'_{\text{uniform}}\|_2$ and $\|\Lambda - \Lambda'_{\text{clustered}}\|_2$) to measure the spectral change.

## C.2 SUMMARY OF DATASETS

The proposed SPAN is evaluated on 25 graph datasets. Specifically, for the *node classification task*, we included Cora, Citeseer, PubMed citation networks (Sen et al., 2008), Wiki-CS hyperlink network (Mernyei & Cangea, 2020), Amazon-Computer and Amazon-Photo co-purchase network (Shchur et al., 2018), and Coauthor-CS network (Shchur et al., 2018). For the *graph classification and regression tasks*, we employed TU biochemical and social networks (Morris et al., 2020), Open Graph Benchmark (OGB) (Hu et al., 2020a) and ZINC (Hu et al., 2020b; Gómez-Bombarelli et al., 2018) chemical molecules, and Protein-Protein Interaction (PPI) biological networks (Hu et al., 2020b; Zitnik & Leskovec, 2017). We summarize the statistics of these datasets and briefly introduce the experiment settings on them.

- A collection of datasets were used to evaluate *node classification* performance in both *unsupervised learning* and *adversarial attack* settings, and Table 5 summarizes the statistics of these datasets. **Cora, Citeseer, PubMed** citation networks (Sen et al., 2008) contain nodes representing documents and edges denoting citation links. The task is to predict the research topic of a document given its bag-of-word representation. **Wiki-CS** hyperlink network (Mernyei & Cangea, 2020) consists of nodes corresponding to Computer Science articles, with edges based on hyperlinks. The task is to predict the branch of the field about the article using its 300-dimension pretrained GloVe word embeddings. **Amazon-Computer, Amazon-Photo** co-purchase networks (Shchur et al., 2018) have nodes being items and edges representing that two items are frequently bought together. Given item reviews as bag-of-word node features, the task is to map items to their respective item category. **Coauthor-CS** network (Shchur et al., 2018) contains node to be authors and edges to be co-author

Table 7: OGB chemical molecular datasets (Hu et al., 2020a) for both graph classification and regression tasks in unsupervised learning setting. The evaluation metric used for regression task is *RMSE*, and for classification is *ROC-AUC*.

| Task Type | Name | #Graph | Avg. #Nodes | Avg. #Edges | #Tasks |
|---|---|---|---|---|---|
| Regression | ogbg-molesol | 1,128 | 13.3 | 13.7 | 1 |
| | ogbg-molipo | 4,200 | 27.0 | 29.5 | 1 |
| | ogbg-molfreesolv | 642 | 8.7 | 8.4 | 1 |
| Classification | ogbg-molbace | 1,513 | 34.1 | 36.9 | 1 |
| | ogbg-molbbbp | 2,039 | 24.1 | 26.0 | 1 |
| | ogbg-molclintox | 1,477 | 26.2 | 27.9 | 2 |
| | ogbg-moltox21 | 7,831 | 18.6 | 19.3 | 12 |
| | ogbg-molsider | 1,427 | 33.6 | 35.4 | 27 |

Table 8: Biological interaction and chemical molecular datasets (Hu et al., 2020b) for graph classification task in transfer learning setting. The evaluation metric is *ROC-AUC*.

| Data Type | Stage | Name | #Graph | Avg. #Nodes | Avg. #Degree |
|---|---|---|---|---|---|
| Protein-Protein Interaction Networks | Pre-training | PPI-306K | 306,925 | 39.82 | 729.62 |
| | Fine-tuning | PPI | 88,000 | 49.35 | 890.77 |
| Chemical Molecules | Pre-training | ZINC-2M | 2,000,000 | 26.62 | 57.72 |
| | Fine-tuning | BBBP | 2,039 | 24.06 | 51.90 |
| | | Tox21 | 7,831 | 18.57 | 38.58 |
| | | SIDER | 1,427 | 33.64 | 70.71 |
| | | ClinTox | 1,477 | 26.15 | 55.76 |
| | | BACE | 1,513 | 34.08 | 73.71 |
| | | HIV | 41,127 | 25.52 | 54.93 |
| | | MUV | 93,087 | 24.23 | 52.55 |
| | | ToxCast | 8,576 | 18.78 | 38.52 |

relationship. Given keywords of each author's papers, the task is to map authors to their respective field of study. All of these datasets are included in the PyG (PyTorch Geometric) library[3].

- Two sets of datasets were used to evaluate *graph prediction* tasks under the *unsupervised learning* setting. **TU Datasets** (Morris et al., 2020) provides a collection of benchmark datasets, and we used several biochemical molecules and social networks for graph classification as summarized in Table 6. The data collection is also included in the PyG library following a 10-fold evaluation data split. We used these datasets for evaluation of the graph classification task in unsupervised learning setting. **Open Graph Benchmark (OGB)** (Hu et al., 2020a) contains datasets for chemical molecular property classification and regression tasks, which are summarized in Table 7. This data collection can be load via the OGB platform [4], and we used its processed format available in PyG library.

- A set of biological and chemical datasets were used to evaluate *graph classification* task under the *transfer learning* setting, summarized in Table 8. Following the transfer learning pipeline in (Hu et al., 2020b), an encoder was first pre-trained on a large biological Protein-Protein Interaction (**PPI**) network or **ZINC** chemical molecule dataset, and then was evaluated on small datasets from the same domains.

## C.3 SUMMARY OF GCL BASELINES

We compared GCL-SPAN against seven self-supervised learning baselines for *node representation learning*, including GRACE (Zhu et al., 2020), its extension GCA (Zhu et al., 2021), BGRL (Thakoor

---

[3]https://pytorch-geometric.readthedocs.io/en/latest/index.html
[4]https://ogb.stanford.edu/docs/graphprop/

et al., 2021), GBT (Bielak et al., 2021), MVGRL (Hassani & Khasahmadi, 2020), GMI (Peng et al., 2020) and DGI (Veličković et al., 2019). Meanwhile, five baselines designed for *graph representation learning* were compared, including InfoGraph (Sun et al., 2019), GraphCL (You et al., 2020a), MVGRL (Hassani & Khasahmadi, 2020), AD-GCL (with fixed regularization weight) (Suresh et al., 2021) and JOAO (v2) (You et al., 2021). In the contrastive objective design of these methods, different contrastive modes are adopted to compare node-level or graph-level representations. We summarize them based on their contrastive modes as follows:

- *Node v.s. node* mode specifies the contrastive examples as node pairs in a local perspective, which focuses on node-level representation learning to serve node prediction tasks. In particular, **GRACE** (Zhu et al., 2020) employs uniformly random edge removing to generate two views, and treats the same node from two views as positive pairs, and all the other nodes as negatives. **GCA** (Zhu et al., 2021) extends GRACE (Zhu et al., 2020) with an adaptive augmentation considering the node centrality. **BGRL** (Thakoor et al., 2021) adopts uniformly random edge removing augmentation and applies a bootstrapping (Grill et al., 2020) framework to avoid collapse without negative sampling. **GBT** (Bielak et al., 2021) uses uniformly random edge removing as graph augmentation and a Barlow-twins (Zbontar et al., 2021) objective to avoid collapse without requiring negative sampling. **GMI** (Peng et al., 2020) maximizes a general form of graphical mutual information defined on both features and edges between nodes in input graph and reconstructed output graph.

- *Graph v.s. node* mode takes graph and node pairs as contrastive examples to decide whether they are from the same graph, which obtains both node- and graph-level representations. In particular, **MVGRL** (Hassani & Khasahmadi, 2020) maximizes the mutual information between the local Laplacian matrix and a global diffusion matrix, which obtains both node-level and graph-level representations that can serve for both node and graph prediction tasks. **DGI** (Veličković et al., 2019) proposes to maximize the mutual information between representations of local nodes and the entire graph, in contrast with a corrupted graph by node shuffling. **InfoGraph** (Sun et al., 2019) aims to maximize the mutual information between the representations of entire graphs and substructures (e.g., nodes, edges and triangles) with different granularity, and it is evaluated on graph-level prediction tasks.

- *Graph v.s. graph* mode treats contrastive examples as graph pairs from a global perspective, which mainly targets on graph-level representation learning for graph prediction tasks. Specifically, **AD-GCL** (Suresh et al., 2021) aims to avoid capturing redundant information during the training by optimizing adversarial graph augmentation strategies in GCL, and designs a trainable non-i.i.d. edge-dropping graph augmentation. **JOAO** (You et al., 2021) adopts a bi-level optimization framework to search the optimal strategy among multiple types of augmentations such as uniform edge or node dropping, subgraph sampling. **GraphCL** (You et al., 2020a) extensively studies graph structure augmentations including random edge removing, node dropping and subgraph sampling.

### C.4 HYPER-PARAMETER SETTING

**Training Configuration.** We summarize the configuration of our GCL framework, including the GNN encoder and training parameters. For node representation learning, we used GCN (Kipf & Welling, 2017) encoder, and set the number of GCN layers to 2, the size of hidden dimension for each layer to 512. The training epoch is 1000. For graph representation learning, we adopted GIN (Xu et al., 2019) encoder with 5 layers, which was concatenated by a readout function that adds node representations for pooling. The embedding size was set to 32 for TU dataset and 300 for OBG dataset. We used 100 training epochs with batch size 32. In all the experiments, we used the Adam optimizer with learning rate $0.001$ and weight decay $10^{-5}$. For data augmentation, we adopted both edge perturbation and feature masking, whose perturbation ratio $\sigma_e$ and $\sigma_f$ were tuned by grid search among $\{0.1, 0.2, 0.3, 0.4, 0.5, 0.6, 0.7, 0.8, 0.9\}$ based on the validation set. Note that in our formulation Eq. 9, the augmentation strength $\epsilon = \sigma_e \cdot m$ where $m$ is the number of edges in the input graph.

**Evaluation Protocol.** When evaluating the quality of learned representations on downstream tasks in an unsupervised setting, we adopted the evaluation protocol proposed in (Suresh et al., 2021). Specifically, based on the representations given by the encoder, we trained and evaluated a Logistic classifier or a Ridge regressor with $L_2$ regularization, whose weight was tuned among $\{0.001, 0.01, 0.1, 1.0, 10.0, 100.0\}$ on the validation set for each dataset.

## C.5 OUT-OF-DOMAIN TRANSFER LEARNING

A more challenging transfer learning setting is proposed in (Qiu et al., 2020), which supports model training on multiple graphs from academic and social networks and transferring to different downstream tasks for other datasets. We further conducted a transfer learning experiment under such setting to demonstrate the out-of-domain transferability of our proposed GCL solution GCL-SPAN. Specifically, we pre-train the encoder on the Yelp dataset (Zeng et al., 2020) which contains 716,847 nodes and 13,954,819 edges; and then the encoder is fine-tuned and evaluated on the US-airport dataset (Ribeiro et al., 2017) for node classification task and three TU social networks (Morris et al., 2020) for graph classification task via 10-fold CV. We sample 80,000 2-hop ego-nets from the Yelp dataset for pre-training, and use node degree as the node feature for all datasets.

Table 9 summarizes the results of downstream classification accuracy. Overall, we can observe satisfactory performance gain using contrastive learning to pre-train the GIN encoder on Yelp dataset under such a cross-domain setting. Meanwhile, our proposed method GCL-SPAN outperforms other contrastive learning methods on three out of four downstream datasets.

Table 9: Node and graph classification performance under *out-of-domain transfer learning* setting. The metric is *accuracy%*. The best and second best results are highlighted with **bold** and underline respectively.

| Dataset | Pre-Train | Yelp | | | |
|---|---|---|---|---|---|
| | Fine-Tune | Node Classification | Graph Classification | | |
| | | US-Airport | COLLAB | RDT-B | IMDB-B |
| No-Pre-Train-GIN | | 62.42±1.27 | 74.82±0.92 | 86.79±2.04 | 71.83±1.93 |
| Baseline | MVGRL | 63.83±0.97 | 74.78±0.84 | 86.24±1.26 | 73.21±1.54 |
| | AD-GCL | – | 75.11±0.70 | **88.72±1.53** | 74.34±1.23 |
| | JOAO | – | 75.35±0.93 | 87.65±1.72 | 75.15±1.67 |
| GCL-SPAN | | **65.21±0.86** | **76.37±0.73** | 88.41±1.12 | **75.89±1.20** |

# D MODEL ANALYSIS

## D.1 EMPIRICALLY RUNNING TIME

Recall that the time complexity of augmentation pre-computation is $\mathcal{O}(TKn^2)$. Table 10 shows the empirical running time (in seconds) comparison between the pre-computation of the spectrum-based augmentation scheme and the follow-up contrastive learning iterations. Specifically, in these experiments, we used $K = 1000$, and $T = 50$ for node classification on four representative datasets with varying node sizes.

Table 10: Empirical running time (in seconds) on four representative node classification datasets with varying node sizes.

| | Cora | Amazon-Photo | Wiki-CS | PubMed |
|---|---|---|---|---|
| #nodes | 2,708 | 7,650 | 11,701 | 19,717 |
| Augmentation pre-computation time ($K = 1000, T = 50$) | 105.4 | 512.6 | 893.2 | 2462.3 |
| Contrastive training time ($epoch = 1000$) | 262.9 | 887.5 | 1052.4 | 1429.2 |

Compared with the time needed for performing contrastive learning, the pre-computation cost is comparable and acceptable. For large-scale graphs with higher time and memory complexity, we can further adopt the widely employed treatments in practice (e.g., ego-nets sampling, augmenting sampled ego-nets and training in batch), as introduced in GCC (Qiu et al., 2020).

We also want to emphasize that our work is the first step towards effective augmentation for graph contrastive learning by exploiting the graph spectral property. As promising performance gain is

observed in our study, the next step is to improve the efficiency where practical treatments for training in large graphs can be applied.

## D.2 INFLUENCE OF CHOOSING EIGENVALUES

**Sensitivity of $K$.** To reduce the time complexity of eigen-decomposition when calculating the spectrum norm $\mathcal{L}_{GS}(\mathbf{\Delta})$, we can approximate the norm by only using the $K$ lowest- and highest-eigenvalues. The time complexity of optimizing the augmentation scheme in Eq. 7 with $T$ iterations is $\mathcal{O}(TKn^2)$. This experiment shows the influence of $K$ to the resulting GCL performance. Since the graphs encountered in the node prediction tasks are much larger than those in graph prediction tasks, we used the node classification datasets in Table 5 to conduct this experiment. Specifically, we test influence of $K$ on four large graphs representing different types of networks: PubMed citation network, Wiki-CS hyperlink network, Amazon-Computers co-purchase network and Coauthor-CS network. We tuned $K$ among $\{50, 100, 200, 500, 1000, 5000\}$ for each of the datasets containing $n \geq 10,000$ nodes. The other components of GCL maintained the same, except the resulting augmentation scheme using spectrum norm with different $K$.

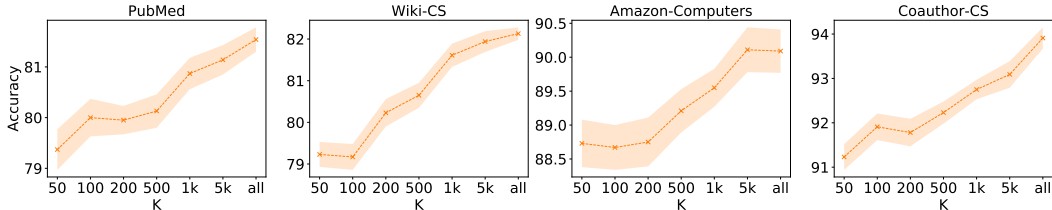

Figure 3: Node classification performance when choosing $K$ lowest- and highest-eigenvalues

Figure 3 demonstrates the performance of GCL-SPAN on the node classification task when different $K$ was used to generate the augmentation scheme. The x-axis denotes the value of $K$ with "all" indicating that all the eigenvalues were used. The performance decreases marginally when we used a smaller $K$, and generally when $K = 1000$ we can still achieve a comparable performance. This suggests that low and high eigenvalues are already quite informative in capturing graph structural properties. Similar phenomenon is also discussed in previous works (Lin et al., 2021): small eigenvalues carry smoothly varying signals (e.g., similar neighbor nodes within the same cluster), while high eigenvalues carry sharply varying signals (e.g., dissimilar nodes from different clusters).

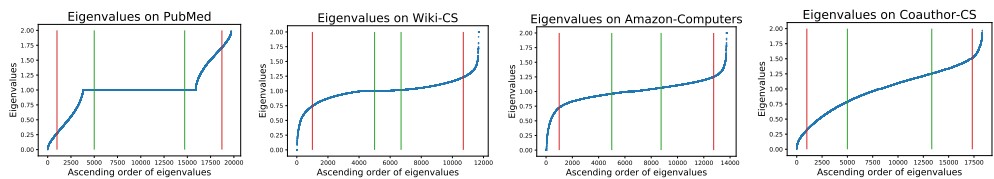

Figure 4: Coverage of eigenvalues when choosing $K = $1k, 5k.

**Coverage of $K$ on Graph Spectrum.** To show the selection of $K$ still covers a moderate range of eigenvalues, we show the distribution of eigenvalues on several real-world graphs in Figure 4. The x-axis denotes the ascending order of $n$ eigenvalues, and y-axis shows the eigenvalue at the corresponding order. Red vertical lines represents $K = 1000$ lowest and highest eigenvalues, while green vertical lines denotes $K = 5000$. We can observe that $K = 1000$ can cover a moderate range of the eigenvalues, and setting larger $K = 5000$ can well cover most of the eigenvalues. Therefore even if we only choose these eigenvalues, we can already achieve empirically satisfactory performance indicated by Figure 3.

## D.3 GAIN OF SPAN ON OTHER GCL FRAMEWORKS

In this experiment, we use an ablation study to evaluate the effectiveness of the graph spectrum guided topology augmentation scheme when applied to different contrastive learning frameworks. We

focus on GCL for node-level representation learning, as this line of work adopts distinct contrastive objectives (e.g., bootstrapping in BGRL, and Barlow twins in GBT) and contrastive modes (e.g., node v.s. node in GRACE, and node v.s. graph in MVGRL), such that we can comprehensively demonstrate the effectiveness of our proposed augmentation in covering a variety of GCL frameworks.

Specifically, we replace the original uniformly random edge removing augmentation in GRACE, BGRL, GBT, and the diffusion matrix based augmentation in MVGRL with the proposed spectrum based augmentation scheme, and use *-SPAN* as suffix to denote them. Note that MVGRL-SPAN is basically GCL-SPAN since it uses the same contrastive objective as in MVGRL such that both node- and graph-level representations are obtained to serve a broader range of downstream tasks.

Table 11: Node classification performance under *unsupervised* setting. We plug the spectrum based augmentation to different GCL frameworks, highlighted with suffix *-SPAN*. The metric is *accuracy%*. The best results are highlighted with **bold**.

| Dataset | Cora | Citeseer | PubMed | Wiki-CS | Amazon-Computer | Amazon-Photo | Coauthor-CS |
|---|---|---|---|---|---|---|---|
| GRACE | 83.33±0.43 | 72.10±0.54 | 78.72±0.13 | 80.14±0.48 | 89.53±0.35 | **92.78±0.30** | 91.12±0.20 |
| **GRACE-SPAN** | **84.21±0.51** | **72.87±0.58** | **79.94±0.22** | **80.63±0.47** | **89.95±0.41** | 92.56±0.34 | **91.98±0.20** |
| BGRL | 83.63±0.38 | 72.52±0.40 | 79.83±0.25 | 79.98±0.13 | **90.34±0.19** | 93.17±0.30 | 93.31±0.13 |
| **BGRL-SPAN** | **84.34±0.42** | **72.73±0.44** | **80.78±0.32** | **81.04±0.22** | 90.12±0.21 | **93.58±0.39** | **93.77±0.21** |
| GBT | 80.24±0.42 | 69.39±0.56 | 78.29±0.43 | 76.65±0.62 | 88.14±0.33 | 92.63±0.44 | 92.95±0.17 |
| **GBT-SPAN** | **82.43±0.51** | **71.12±0.48** | **80.05±0.49** | **78.89±0.54** | **89.04±0.43** | **92.78±0.43** | 92.95±0.37 |
| MVGRL | 85.16±0.52 | 72.14±1.35 | 80.13±0.84 | 77.52±0.08 | 87.52±0.11 | 91.74±0.07 | 92.11±0.12 |
| **MVGRL-SPAN** | **85.86±0.57** | **72.76±0.63** | **81.54±0.24** | **82.13±0.15** | **90.09±0.32** | **93.52±0.26** | **93.91±0.24** |

Table 11 shows the results of plugging our augmentation scheme on four types of GCL frameworks. We can observe that our augmentation scheme does not depend on a particular contrastive objective, but brings a clear performance gain across different GCL frameworks. Intuitively, our augmentation captures the essential structural properties by perturbing edges that cause large spectral change. Therefore, no matter what contrastive objective or mode is used, maximizing the correspondence of two views encourages the encoder to ignore the information carried by such sensitive edges. This demonstrates the importance of studying graph spectral properties for graph augmentation.

### D.4 ANALYSIS OF PERTURBATION STRENGHTH

The value of $\epsilon$ controls the perturbation strength when generating augmented graphs. A larger $\epsilon$ value indicates that more edges will be dropped/added. Specifically, the optimized scheme $\Delta_1, \Delta_2$ constrained by $\epsilon$ will in expectation perturb $\epsilon = \sigma_e \times m$ edges in the augmented views, where $m$ is the number of edges in the input graph and $\sigma_e$ is the perturbation rate. To analyze the effect of perturbation strength $\epsilon$, we tune $\sigma_e = \epsilon/m = \{0.1, 0.2, \ldots, 0.9\}$, and compare the proposed spectral augmentation with uniformly random edge augmentation on the same GCL instantiation shown in Figure 2. The performance comparison is conducted under unsupervised node classification task.

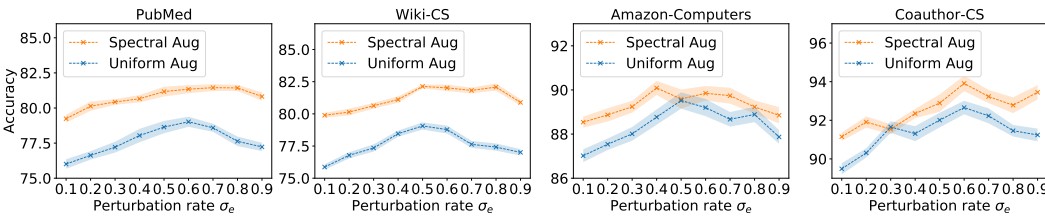

Figure 5: Node classification performance when tuning perturbation ratio $\sigma_e = \epsilon/m$.

In Figure 5, the x-axis shows the perturbation rate $\sigma_e$, which denotes the perturbation strength $\epsilon$ normalized by total edge number of the graph (e.g. $\sigma_e = \epsilon/m$). The performance of spectral augmentation in general is more stable under different perturbation strength, compared with the uniformly random augmentation. This demonstrates that our proposed augmentation can better preserve structural invariance by assigning larger perturbation probability to sensitive edges.

D.5    GAIN OF SPAN COMPARED WITH OTHER AUGMENTATION SCHEMES

We provide additional experiments by fixing the contrastive learning framework, but varying the augmentation schemes, including two existing edge perturbation schemes (e.g. the widely used uniformly random edge perturbation, and the diffusion augmentation proposed in MVGRL (Hassani & Khasahmadi, 2020)), as well as multiple proposed schemes in this paper (e.g., single-way, double-way and opposite-direction schemes). We compare these schemes with our GCL instantiation in Figure 2 on unsupervised node classification tasks.

Table 12: Node classification performance under *unsupervised* setting. We plug different augmentation schemes to the GCL instantiation in Figure 2. The metric is *accuracy%*.

| Dataset | Cora | Amazon-Photo | Wiki-CS | PubMed |
|---|---|---|---|---|
| Uniformly random | 83.87±0.56 | 91.78±0.27 | 79.06±0.31 | 79.02±0.58 |
| Diffusion augmentation | 85.16±0.52 | 91.74 ±0.07 | 77.52±0.08 | 80.13±0.84 |
| Singe-way scheme SPAN$_{single}$ | 84.38±0.42 | 92.28±0.29 | 80.76±0.34 | 80.65±0.31 |
| Double-way scheme SPAN$_{double}$ | 84.93±0.64 | 91.81±0.33 | 81.08±0.47 | 79.43±0.36 |
| Opposite-direction scheme SPAN$_{opposite}$ | 85.86±0.57 | 93.52±0.26 | 82.13±0.15 | 81.54±0.24 |

Table 12 shows the comparison. The opposite-direction scheme is shown to be the most effective augmentation, especially when comparing with the widely used uniformly random augmentation. The effectiveness of the diffusion augmentation depends on the datasets: it is not even as good as the uniformly random method on Wiki-CS. The single-way scheme only considers spectrum on one branch, thus has limited improvement, and the double-way scheme can not well maximize the spectral difference of two views due to suboptimal solutions to the optimization problem.

D.6    SPATIAL BEHAVIOR OF SPECTRAL AUGMENTATION SPAN

**Case Study on Random Geometric Graph.** To intuitively show the spatial change caused by spectral augmentation, we visualize a case study on a random geometric graph in Figure 6. Figure 6a draws the original graph. Figure 6b shows the perturbation probability obtained by maximizing $\mathcal{L}_{GS}(\boldsymbol{\Delta})$. Figure 6c illustrates the perturbation probability obtained by minimizing $\mathcal{L}_{GS}(\boldsymbol{\Delta})$.

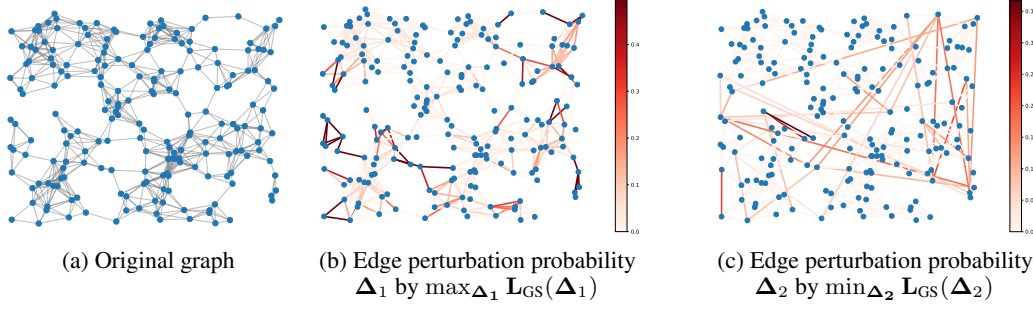

(a) Original graph     (b) Edge perturbation probability $\boldsymbol{\Delta}_1$ by $\max_{\boldsymbol{\Delta_1}} \mathbf{L}_{GS}(\boldsymbol{\Delta}_1)$     (c) Edge perturbation probability $\boldsymbol{\Delta}_2$ by $\min_{\boldsymbol{\Delta_2}} \mathbf{L}_{GS}(\boldsymbol{\Delta}_2)$

Figure 6: A case study of the spectrum-guided augmentation scheme on a random geometric graph.

We can observe that maximizing the spectral norm assigns larger probability to remove edges bridging clusters such that the clustering effect becomes more obvious, while minimizing the spectral norm tends to add edges connecting clusters such that the clustering effect is blurred. Intuitively, such an augmentation perturbs these spurious edges that can easily affects the structural property (e.g. clustering) to preserve structural invariant, and the information about these edges are disentangled and minimized in the learned representations by contrastive learning.

**Theorectical Justification.** We also provide a theoretical justification which reveals the interplay of spectral change and spatial change. As derived in Theorem 2 of (Bojchevski & Günnemann, 2019), given an edge flip $\Delta w_{ij} = (1 - 2A_{ij}) \in \{-1, 1\}$ between node $i$ and $j$ (e.g. if $\Delta w_{ij} = 1$, adding edge $(i, j)$; otherwise removing edge $(i, j)$), the $k$-th eigenvalue is changed as $\lambda'_k = \lambda_k + \Delta\lambda_k$. $\Delta\lambda_k$

can be approximated by:

$$\Delta\lambda_k = \Delta w_{ij}(2u_{ki} \cdot u_{kj} - \lambda_k(u_{ki}^2 + u_{kj}^2)) \tag{14}$$

where $u_k$ is the $k$-th eigenvector corresponding to the eigenvalue $\lambda_k$. If we only focus on the magnitude of eigenvalue change, we can obtain:

$$|\Delta\lambda_k| = |(u_{ki} - u_{kj})^2 + (\lambda_k - 1)(u_{ki}^2 + u_{kj}^2)| \tag{15}$$

**Remarks**. Since the eigenvectors are normalized, we can treat $(u_{ki}^2 + u_{kj}^2)$ as a constant as it is a relatively stable value. Based on the theory in spectral clustering (Ng et al., 2001), if the eigenvectors of node $i$ and node $j$ have a larger difference (i.e., $\|u_{.,i} - u_{.,j}\|_2$ is large), these two nodes should belong to different clusters. The first term in Eq. 15 suggests a larger eigenvalue change, if $u_{ki}$ and $u_{kj}$ have a larger difference. Therefore, flipping the edge between nodes from different clusters (thus with a larger $(u_{ki} - u_{kj})^2$) results in a larger spectral change. The second term suggests that such an effect becomes more obvious for eigenvalues close to 0 or 2.

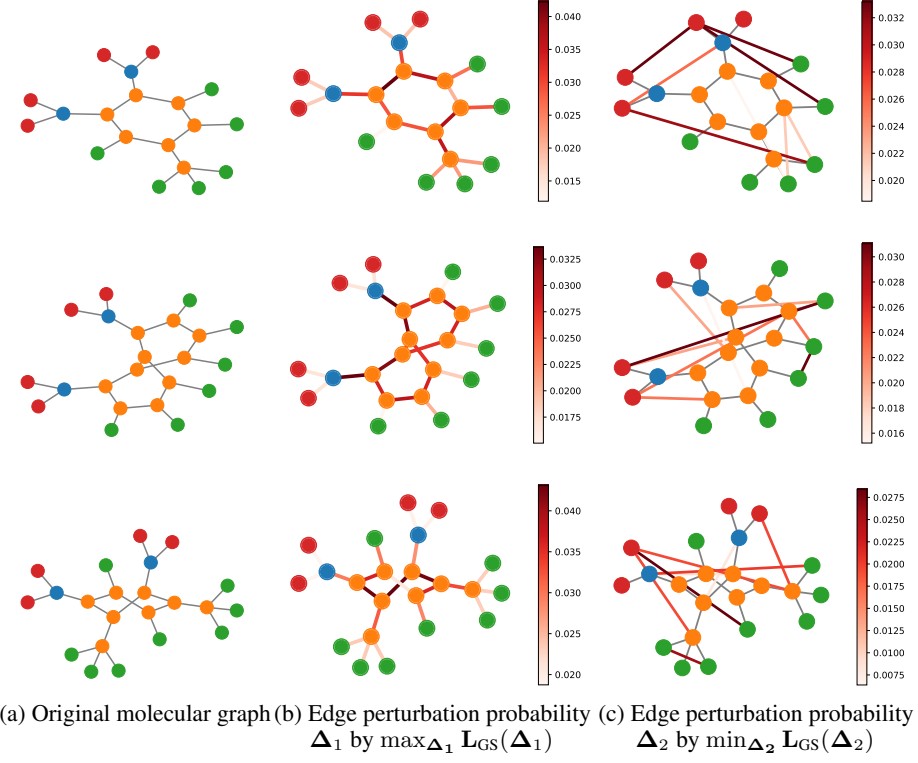

(a) Original molecular graph (b) Edge perturbation probability $\mathbf{\Delta}_1$ by $\max_{\mathbf{\Delta_1}} \mathbf{L}_{\text{GS}}(\mathbf{\Delta}_1)$ (c) Edge perturbation probability $\mathbf{\Delta}_2$ by $\min_{\mathbf{\Delta_2}} \mathbf{L}_{\text{GS}}(\mathbf{\Delta}_2)$

Figure 7: A case study of the spectrum-guided augmentation scheme on MUTAG molecular graphs.

**Case Study on MUTAG Molecular Graphs.** To further investigate the effect of proposed augmentation on real-world graphs, we also visualize the augmentation scheme (perturbation probability matrices $\Delta_1, \Delta_2$) on several MUTAG molecular graphs in Figure 7. Each row shows one molecular graph with mutagenicity. The first column illustrates the original graph. The second column shows the perturbation probability obtained by maximizing $\mathcal{L}_{\text{GS}}(\mathbf{\Delta})$. The last column illustrates the perturbation probability obtained by minimizing $\mathcal{L}_{\text{GS}}(\mathbf{\Delta})$. Node color represents the atom type: yellow, blue, red and green denotes carbon, nitrogen, oxygen, and hydrogen, respectively.

We can observe that the augmentation scheme assigns high perturbation probability to the edges across different chemical groups. For example, there is a higher probability to drop/add edges between $NO_2$ and carbon rings. By contrastive learning, the information about edges blurring the boundary between $NO_2$ and other atoms is minimized. Therefore, the key functional group $NO_2$ is preserved, which are shown important for predicting the molecule's mutagenicity (Debnath et al., 1991; Ying et al., 2019; Luo et al., 2020).

### D.7 The Convergence of Optimizing the Spectral Augmentation

In this section, we show how the graph spectrum norm changes as the augmentation optimization proceeds following Eq. 7. To better show the relative change of graph spectrum compared with the original graph, we calculate $\mathcal{L}_{GS}(\boldsymbol{\Delta})$ normalized by the spectrum norm of the original graph, that is, $\mathcal{L}_{GS}(\boldsymbol{\Delta})/\mathcal{L}_{GS}(\mathbf{0}) = \|eig(Lap(\mathbf{A} + \mathbf{C} \circ \boldsymbol{\Delta}))\|_2^2/\|eig(Lap(\mathbf{A}))\|_2^2$. The value of normalized $\mathcal{L}_{GS}(\boldsymbol{\Delta})/\mathcal{L}_{GS}(\mathbf{0})$ is reported in Figure 8.

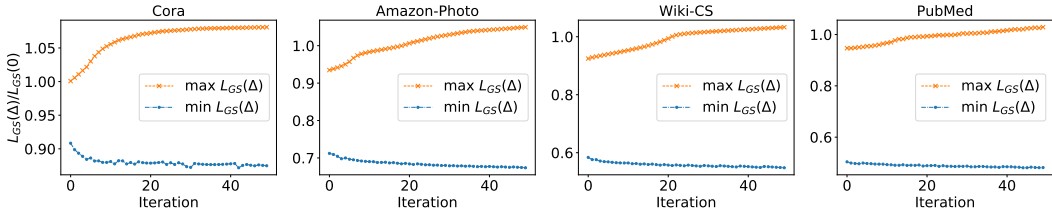

Figure 8: The value of relative spectral change $\mathcal{L}_{GS}(\boldsymbol{\Delta})/\mathcal{L}_{GS}(\mathbf{0})$ when maximizing the spectrum norm (orange) or minimizing it (blue) via Eq. 7

Based on the graph spectral theory, the eigenvalues are bounded within $[0, 2]$, thus the L2-norm of the graph spectrum is also bounded by the total number of nodes, For example, the values of the original spectrum norm $\mathcal{L}_{GS}(\mathbf{0})$ for Cora, Amazon-Photo, Wiki-CS and PubMed are $24.88, 22.13, 31.79$, and $65.26$ respectively. From Figure 8, we can observe that maximizing $\mathcal{L}_{GS}(\boldsymbol{\Delta})$ indeed results in an a larger spectrum norm compared with the original graph (i.e. $\mathcal{L}_{GS}(\boldsymbol{\Delta})/\mathcal{L}_{GS}(\mathbf{0}) > 1$), while minimizing it achieves a smaller spectrum norm (i.e. $\mathcal{L}_{GS}(\boldsymbol{\Delta})/\mathcal{L}_{GS}(\mathbf{0}) < 1$).

## E Extension to Support More Augmentation Types

Table 13: Graph classification performance using node dropping augmentation schemes on the GCL instantiation in Figure 2. The metric is *accuracy%*.

| Dataset | Biochemical Molecules | | Social Networks | |
|---|---|---|---|---|
| | NCI1 | PROTEINS | COLLAB | IMDB-M |
| Uniformly random node dropping | $69.27 \pm 0.86$ | $73.40 \pm 0.74$ | $75.19 \pm 0.67$ | $53.04 \pm 0.63$ |
| Spectral guided node dropping | $70.96 \pm 0.77$ | $74.51 \pm 0.58$ | $75.65 \pm 0.55$ | $53.77 \pm 0.61$ |

The proposed principle from the graph spectral perspective can also be extended to node dropping augmentation. We design a soft node dropping scheme, which assigns a dropping probability to each node. Different from the edge perturbation scheme, node dropping is sampled from a Bernoulli distribution $\mathcal{B}(\mathbf{p})$, where $\mathbf{p} \in [0, 1]^n$. We can sample a node dropping vector $\mathbf{d} \in \{0, 1\}^n$, where $d_i \sim \mathcal{B}(\mathbf{p})$ indicates whether to drop the node $i$, and the node is dropped if $d_i = 1$.

Dropping a node is equivalent to removing all the edges connected to this node. Therefore, we can extend the operation of node dropping to edge removal. The node dropping probability $\mathbf{p}$ implies the following edge removing probability matrix $\mathbf{P}$:

$$\mathbf{P} = \frac{\mathbf{p} \cdot \mathbf{1}^\top + (\mathbf{p} \cdot \mathbf{1}^\top)^\top}{2} \tag{16}$$

where $\mathbf{1}$ is an all-one vector with dimension $n$. The node dropping based augmentation scheme can be then obtained by:

$$T(\mathbf{A}) = \mathbf{A} + (-\mathbf{A}) \circ \mathbf{P} \tag{17}$$

where $\circ$ is an element-wise product. To optimize the node dropping probability vector $\mathbf{p}$, we can follow Eq. 5 by replacing the edge perturbation scheme $\mathbf{A} + \mathbf{C} \circ \boldsymbol{\Delta}$ with the node dropping scheme:

$$\max_{\mathbf{p} \in \mathcal{S}} \|eig(Lap(\mathbf{A} + (-\mathbf{A}) \circ \mathbf{P})) - eig(Lap(\mathbf{A}))\|_2^2 \tag{18}$$

where $\mathcal{S} = \{\mathbf{s}|\mathbf{s} \in [0,1]^n, \|\mathbf{s}\|_1 \leq \epsilon\}$ and $\epsilon$ controls the node perturbation strength. Following similar optimization step, the node dropping probability vector $\mathbf{p}$ can be obtained. Augmented views are then sampled from the optimized probability $\mathbf{p}$ to drop nodes.

We empirically compared the spectrum guided node dropping augmentation with uniformly random node dropping strategy. Table 13 reports their prediction accuracy on four graph classification datasets. We can still observe that the spectral guided node dropping augmentation achieves better performance, which demonstrates the applicability of our proposed principle on both edge and node augmentation. To enable more general topology augmentation, the spectral distance of two graphs can be further extended from L2 distance to distribution divergence, such as Wasserstein distance, to capture the distributional change of graph spectrum.

