# OpenReview forum: "Spectral Augmentation for Self-Supervised Learning on Graphs"
_ICLR.cc/2023/Conference — ICLR 2023 notable top 25%_

### Official Review · Reviewer_eFML · 2022-10-25

**Confidence:** 3
**Correctness:** 4
**Technical Novelty And Significance:** 3
**Empirical Novelty And Significance:** 3
**Recommendation:** 8

**Clarity, Quality, Novelty And Reproducibility:**

This paper is well written, and the proposed idea is very novel. Given that the implementation is not submitted as supplementary material, the reproducibility of work is not guaranteed although the implementation details are well provided for most parts.

**Strength And Weaknesses:**

Strength
- The proposed idea of spectral augmentation is novel and technically valid.
- The extensive experiments support the purpose of the spectral augmentation well.

Weaknesses + Questions
- Although spectral augmentation shows improved performance on many graph learning experiments, the way of changing the graph spectrum is somewhat counter-intuitive since the large changes in the graph spectrum imply large topological changes in the original graph. In that sense, it would be good to provide some qualitative examples of how the graph is augmented after spectral augmentation. for example, how do molecular graphs change after augmentation? do they still keep the important functional groups or not?
- The selection of epsilon sounds important. It would be good to have some additional experiments on the analysis of the hyperparameter.
- In experiments, the k-lowest and *highest* eigenvectors are used to augment. We often assume that the highest eigenvectors contain noisy information, and I wonder about the reason behind using the highest eigenvectors.
- Although the eigendecomposition is only performed once for each graph, a graph such as Pubmed still requires a huge amount of time to compute. Will there be a huge difference if one employs some approximation of eigendecomposition?

**Summary Of The Paper:**

This paper proposes a new way to augment the graph-structured data through the analysis of the graph spectrum. Based on the observation that the augmentation changing graph structure dramatically can improve the performance of graph learning, the authors propose a way to find the edges that can dramatically change the spectrum of a graph. The modified graphs are used to augment graph datasets. The experiments are conducted on various datasets for node-level and graph-level classifications. The proposed method induces a significant performance improvement in most cases.


**Summary Of The Review:**

The paper is well-written and easy to follow. The technical details are well provided. The extensive amount of experiments well supports the main claim of the paper. Overall, I lean towards acceptance of this paper.

---

> ### Author Response · Authors · 2022-11-14
> **Response to Reviewer eFML**
>
> We thank the reviewer for the positive comments on our work and valuable suggestions on the experiment and approximation design. Based on the suggestions, we have included a qualitative study on molecular graphs, the ablation study of augmentation strength $\epsilon$, and explanations of the eigendecomposition approximation.
>
> **[Q1]**: The way of changing the graph spectrum is somewhat counter-intuitive, thus it would be good to provide some **qualitative examples** of how the graph is augmented after spectral augmentation. For example, how do molecular graphs change after augmentation, and do they still keep the important functional groups?
>
> **[A1]**: We used an example on **random geometric graph** in Appendix D.5 to demonstrate the spatial behavior (what types of edges are added/removed) of the spectral augmentation. To further elaborate on the effect of proposed augmentation on real-world graphs, as suggested by the reviewer, we also visualize the augmentation scheme (perturbation probability matrices $\Delta_1, \Delta_2$) on several **MUTAG molecular graphs** in **Figure 7, Appendix D.6 in the updated paper**.
>
> In Figure 7, each row shows one molecular graph with mutagenicity. The first column illustrates the original graph. The second column shows the perturbation probability obtained by maximizing $\mathcal{L}_\text{GS}(\mathbf{\Delta})$. The last column illustrates the perturbation probability obtained by minimizing $\mathcal{L}_\text{GS}(\mathbf{\Delta})$. Node color represents the atom type: yellow, blue, red, and green denotes carbon, nitrogen, oxygen, and hydrogen, respectively.
>
> We can observe that the augmentation scheme assigns high perturbation probability to the edges across different chemical groups. For example, there is a high probability to drop/add edges between $NO_2$ and carbon rings. By contrastive learning, the edges blurring the boundary between $NO_2$ and other atoms are treated as noise, and the information about these edges is eliminated. Therefore, the key functional group $NO_2$ is preserved, which is known to be important for predicting the molecule’s mutagenicity [1, 2, 3].
>
> [1] Debnath, Asim Kumar, et al. "Structure-activity relationship of mutagenic aromatic and heteroaromatic nitro compounds. correlation with molecular orbital energies and hydrophobicity." Journal of medicinal chemistry 34.2 (1991): 786-797.
>
> [2] Ying, Zhitao, et al. "Gnnexplainer: Generating explanations for graph neural networks." Advances in neural information processing systems 32 (2019).
>
> [3] Luo, Dongsheng, et al. "Parameterized explainer for graph neural network." Advances in neural information processing systems 33 (2020): 19620-19631.
>
> **[Q2]**: Additional experiments on the analysis of the hyperparameter $\epsilon$.
>
> **[A2]**: Please find our answer to CQ2 in the general response to all reviewers.
>
> **[Q3]**: Since we often assume that the highest eigenvectors contain noisy information, what is the reason behind using the highest eigenvectors?
>
> **[A3]**: Please find our answer to CQ3 in the general response to all reviewers.
>
> **[Q4]**: Eigendecomposition can still be time consuming for large graphs, though it is only performed once. Will there be a huge difference if one employs an approximation of eigendecomposition?
>
> **[A4]**: To effectively compute the eigenvalues and their gradients, in our implementation, we indeed already adopted the efficient approximations of eigendecomposition supported by PyTorch through the LAPACK linear algebra package [1], which is the state-of-the-art Lanczos (or generally Arnoldi) eigensolver. It provides power iteration methods to find the top/bottom $K$ eigenvalues/eigenvectors with guaranteed convergence [1], which reduces the time complexity from $\mathcal{n^3}$ to $\mathcal{Kn^2}$ with n denoting the node number. Our experiments based on such implementation show the effectiveness of our method when eigendecomposition is computed by approximated iterative methods in practice.
>
> We also reported the empirical running time of the augmentation scheme pre-computation in Appendix D.1: compared with the time needed for performing contrastive learning, the pre-computation cost is comparable and acceptable. While this work suggests a new direction to improve augmentation by graph spectrum and observes promising performance, we leave it as our future work to further improve its efficiency by applying more practical treatments (e.g., ego-nets sampling, augmenting sampled ego-nets and training in batch) for large-scale graphs with higher time and memory complexity.

---

### Official Review · Reviewer_2d5T · 2022-10-25

**Confidence:** 4
**Correctness:** 2
**Technical Novelty And Significance:** 3
**Empirical Novelty And Significance:** 3
**Recommendation:** 6

**Clarity, Quality, Novelty And Reproducibility:**

Clarity: The paper is, in general, very well written. The only misgiving on this front is that certain key concepts are not clearly defined and detailed arguments and explanations for some of the key developments in the methodology are missing.

Quality: The paper reports the results of carefully conducted experiments. The main concern with regard to quality is that several claims emerge during the methodology that are not well-supported.

Novelty: The proposed method is novel and represents an interesting and effective approach.

Reproducibility: the paper makes a good effort to provide most of the experimental details. I do not believe that code was provided or that there was a link to code, so this imposes limits on the reproducibility. Even with the detailed information provided, I believe that it would be very challenging to replicate the results.

**Strength And Weaknesses:**

 Strengths:
1.    The approach for generating topology augmentations for GCL by perturbing edges in order to maximize the spectral change is novel.
2.     The experimental analysis is extensive, spanning multiple tasks and including numerous datasets and baselines.
4.     The appendices contain additional experiments that provide considerably more insight into the behaviour of the proposed approach and what aspects lead to the observed performance improvements.

Weaknesses:
1. Some key terms are vaguely defined. “Structural invariance” is defined as “preserv[ing] the essential structural properties” but it is not at all clear what these structural properties are. As a result, some claims in the paper are not well-supported. For example, there does not appear to be genuine support for the claim that “the information about the edges causing structural instability is minimized in the learned representations”.

2.	The connection between structural invariance and spectral invariance is not solid. Yes, the spectrum is strongly related to many structural properties such as clustering, but there is not sufficient evidence here to state that “perturbing graph spectrum controls the change of structural properties”, at least not in the sense that there is a well-understood “control” of the change that is being induced.

3.	The connection from equation (4) to equation (5) is not well-motivated. The initial argument is that a perturbation should be made to maximize the spectral change from the original graph. Then it becomes maximizing the spectral discrepancy between the two views. Finally it becomes maximizing and minimizing the spectral norms. The paper does not clearly explain why the last step aligns with the initial motivation.

4.	The computational cost of the proposed approach is high due to the eigendecomposition. To address this, the authors focus on the K lowest and highest eigenvalues. This takes the optimization task further away again from the initial motivation. The paper offers the vague argument that these “are the most informative suggested by spectral graph theory”, but there are no citations to support this claim, nor any clarification to explain what is meant by “most informative”. It is good that experimental results are included ot explore the impact of varying the number of included eigenvalues, but my concern is that by this point there has been a very significant migration from the initial motivation of preserving structural properties.

Minor point: was there a check whether the normality assumption of the t-test is satisfied?

**Summary Of The Paper:**

This paper proposes a topology augmentation method for graph contrastive learning which explores invariance of graphs from the spectral perspective. To realize the spectral invariance, the paper aims to identify sensitive edges whose perturbation leads to a large spectral difference. The conjecture is that the GNN encoder then focuses more on invariant spectral components. Numerical experiments illustrate the improvement in the performance of unsupervised representation learning, the generalization capability in transfer learning and induced robustness to adversarial attacks.

**Summary Of The Review:**

The paper provides a novel augmentation approach for graph self-supervised learning. Extensive and carefully conducted experiments provide compelling evidence that the proposed method is effective for a range of tasks. The main weaknesses of the paper are  the absence of concrete definitions for key concepts such as "structural invariance", the somewhat weak linkage to "spectral invariance" and then the incorporation of methodological developments and approximations that lead to a final procedure that is relatively far from the initial motivation.

---

> ### Author Response · Authors · 2022-11-14
> **Response to Reviewer 2d5T**
>
> We thank the reviewer for the constructive suggestions to clarify important arguments and strengthen the connection to the initial motivation.
>
> **[Q1]**: “Structure invariance” is defined as “preserving the essential structural properties”, but it is not clear what these structural properties are.
>
> **[A1]**: The structural properties generally reflect the topological information about the graph, including cluster, connectivity, diameter etc, which can be well captured by graph spectrum, as discussed in Appendix A. Structural properties are important to many downstream tasks. For example, clustering reflects how nodes are grouped, which helps node classification tasks (e.g. community detection). We have added corresponding explanations in **Section 4 in the revised paper**.
>
> **[Q2]**: There is no genuine support for the claim that “the information about the edge causing structural instability is minimized in the learned representations”.
>
> **[A2]**: This effect is achieved by the interplay between the opposite-direction augmentation and the contrastive loss: 1) during the augmentation stage, taking the case study in Appendix D.6 and the clustering property as an intuitive example, increasing the spectrum norm leads to removing edges that bridge different node clusters and the clustering effect becomes more obvious, while decreasing the spectrum norm tends to add edges connecting different node clusters and the clustering effect is blurred; 2) then when calculating the contrastive loss to contrast these two opposite augmentations, the edges easily causing structural changes are regarded as noise, and the information related to these edges will be eliminated and disentangled from the learned representations. We have further explained how such results can be obtained in **Section 4 in the revised paper**.
>
> **[Q3]**: The connection between structural invariance and spectral invariance is not valid. There is no sufficient evidence to state that “perturbing graph spectrum controls the change of structural properties”, at least not in the sense that there is a well-understood “control” of the change that is being induced.
>
> **[A3]**: Since the graph spectrum is a comprehensive manifestation of structural properties of the graph, as discussed in Appendix A, structural properties’ change will lead to spectral change. Thus spectral invariance implies structural invariance: if some edges are robust and their perturbation can not lead to a large spectral change (spectral invariance), the structural properties should be not largely changed by changing those edges (structural invariance). Therefore, we can use the spectral invariance as a proxy to capture structural invariance.
>
> The interplay between spectral change and structural property change is studied in Appendix D.6, which takes the clustering property as an example. We can clearly observe that the edge flips between different clusters causing large clustering change leads to large spectral change.
>
> To further avoid confusion, we have rephrased the original statement as “structural properties change leads to spectral change, and spectral invariance implies the structural invariance” in the updated paper.
>
> **[Q4]**: The connection between Eq. (4) and (5) is not well-motivated. Especially, why the last step aligns with the initial motivation.
>
> **[A4]**: The initial motivation is to assign larger perturbation probability to the sensitive edges that can easily disturb the graph spectrum. We elaborate how Eq. (5) is derived from Eq. (3):
>
> - Eq. (3) achieves this goal via a single-branch augmentation framework, where one branch adopts the proposed augmentation while the other branch uses the original graph.
> - Eq. (4) is a more general form of Eq. (3) (e.g. by setting $\Delta_2=0$ in Eq. (3), the solution of Eq. (3) is covered by Eq. (4)), and Eq. (4) achieves the same goal for a two-branch augmentation framework where both branches adopt augmentations.
> - Eq. (5) is a lower bound of Eq. (4). Since directly optimizing Eq. (4) is hard, we appeal to the **triangle inequality** to instead maximize its lower bound: Eq. (4) $\geq$ $\max_{\mathbf{\Delta}_1, \mathbf{\Delta}_2\in\mathcal{S}}$ $|| \text{eig}(\text{Lap}(\mathbf{A+C\circ \Delta}_1)) ||^2_2$ $- || \text{eig}(\text{Lap}(\mathbf{A+C\circ \Delta}_2)) ||^2_2$. And in the lower bound, the optimization of $\mathbf{\Delta}_1$ and $\mathbf{\Delta}$ can be independently conducted, which equivalently leads to Eq. (5).
>
> We have elaborated and strengthened the connection of Eq. (4) and (5) in **Section 5 in the revised paper**.
>
> **[Q5]**: The paper should clarify why the $K$ lowest and highest eigenvalues are the most informative, and what is meant by “most informative”. This could be a significant migration from the initial motivation of preserving structural properties.
>
> **[A5]**: Please find our answer to CQ3 in the general response to all reviewers.

---

> > ### Comment · Reviewer_2d5T · 2022-11-17
> > **Acknowledgement of the authors' response**
> >
> > I appreciate the thoroughness of the response provided by the authors. While some of the responses have helped to clarify issues and resolve concerns, in some cases doubts remain.
> > (1)	I was hoping for a concrete definition of “structural invariance”, preferably mathematical in nature. The term “invariance” has a relatively strict meaning – reflecting that a property does not change after application of a transformation. In this response, and in the revised paper, the discussion is still very vague. It is not clear which property is being preserved and which transformation is being applied. For example, there is the phrase “the invariance should preserve essential structural properties”, but there is never an identification of what the essential structural properties actually are. Phrases like “Structural properties generally reflect the topological information about the graph” are vague – what does “generally reflect” mean? The expression “like clustering” is not helpful (especially since “clustering” is not mathematically defined). Many readers will, of course, be familiar with examples of graph structural properties. But if there is no specification of exactly which structural properties are being considered, then the concept of “structural invariance” is inadequately defined, and this concept is at the core of the paper, with an entire section devoted to it.
> > (2)	There is a similar problem with “spectral invariance”. There is the sentence “We define spectral invariance as the spectral components that are robust to small edge perturbations” – it is hard to see how this is a definition of any form of “invariance”.
> > (3)	With these definitions not being clear, it’s difficult to assess what the claim “spectral invariance implies the structural invariance” actually means. Once that is established, it can be debated as to whether it is true and whether it is demonstrated to be so in the paper.

---

> > > ### Author Response · Authors · 2022-11-18
> > > **Response to Reviewer 2d5T**
> > >
> > > We thank the reviewer for further explaining the concerns about several concepts, which greatly helped us improve the clarity of our paper. We now follow the reviewer’s suggestion to elaborate them with mathematical definitions. The changes are reflected in **Section 4 in the revised paper**.
> > >
> > > **[Round2-Q1]**: What is the concrete definition of “structural invariance” and “spectral invariance”?
> > >
> > > **[Round2-A2]**: By **structural invariance**, we actually mean structural-perturbation invariance, which is defined as the invariance of the GCL encoder’s output when perturbing a constrained number of edges that cause large changes to the structural properties of the input graph:
> > >
> > > $\mathcal{L}_\text{GCL}$   $(\mathbf{A}, t(\mathbf{A}), \Theta)$ $\leq \sigma$,
> > >
> > > s.t.  $t(\mathbf{A})=\text{argmax}_{\|\mathbf{A}-t(\mathbf{A})\|_1\leq \epsilon} \mathcal{D}(p(\mathbf{A}), p(t(\mathbf{A})))$
> > >
> > > where $\Theta$ is the encoder, and $\mathcal{L}_\text{GCL}$ measures the dissimilarity of the encoder output on the original graph (e.g. $\mathbf{A}$) and the augmented graph  $t(\mathbf{A})$. $\mathcal{D(\cdot, \cdot)}$ is a distance metric, and $p(\cdot)$ is a vector-valued function of a graph's structural properties, such as the diameter of the graph, whether the graph is connected, the clustering coefficient, etc. One may focus on particular properties when designing the function $p(\cdot)$ and search for the optimal $t(\mathbf{A})$. However, directly realizing structural invariance defined by a concrete $p(\cdot)$ has clear limitations: 1) the calculation of graph properties is usually non-differentiable w.r.t. edges, thus optimizing $t(\mathbf{A})$ is nontrivial; 2) domain knowledge might be required to select specific properties, and characterizing multiple structural properties simultaneously further complicates the problem.
> > >
> > > In this paper, we propose a general principle of structural invariance measured by the spectral distance, and do not specifically constrain the selection of properties. More specifically, since the graph spectrum is a comprehensive manifestation of structural properties of a graph as shown in Appendix A, we consider **spectral(-perturbation) invariance** as a delegate, which requires the encoder’s output to stay similar when perturbing edges that cause large changes to the graph spectrum. Formally, it is defined as:
> > >
> > > $\mathcal{L}_\text{GCL}(\mathbf{A}, t(\mathbf{A}), \Theta)\leq \sigma,$
> > >
> > > $\text{s.t. } t(\mathbf{A})=\text{argmax}_{\|\mathbf{A}-t(\mathbf{A})\|_1\leq \epsilon} \mathcal{D}(\text{eig}(\text{Lap}(\mathbf{A})), \text{eig}(\text{Lap}(t(\mathbf{A}))))$
> > >
> > > where the distance between two graph structures is measured on the graph spectrum.
> > >
> > > Hope the above clarifies our key insight.
> > >
> > > **[Round2-Q2]**: What does the claim “spectral invariance implies structural invariance” means? And is this claim demonstrated true?
> > >
> > > **[Round2-A2]**: Given the definition of structural and spectral invariance, this claim suggests that if we train an encoder that satisfies spectral invariance, this encoder is also invariant to edge perturbations that cause large changes to structural properties. Therefore when applying this encoder to a downstream task, it should be robust to such structural perturbations.
> > >
> > > This claim can be supported by our adversarial robustness evaluation in Section 7.3, where DICE [1] is an attack method that adds inter-cluster edges and removes intra-cluster edges to disturb the clustering property of a graph. We can observe that the encoder trained by our proposed method, i.e., preserving spectral invariance, is robust to such an attack, which indicates its invariance to structural perturbation.
> > >
> > > [1] Waniek, Marcin, et al. "Hiding individuals and communities in a social network." Nature Human Behaviour 2.2 (2018): 139-147.

---

### Official Review · Reviewer_fWm4 · 2022-11-04

**Confidence:** 5
**Clarity, Quality, Novelty And Reproducibility:** 1. I think the clarity is good, and I…
**Correctness:** 2
**Technical Novelty And Significance:** 2
**Empirical Novelty And Significance:** 3
**Recommendation:** 6

**Strength And Weaknesses:**

#### Strengths
1. The description and formulation of the proposed algorithm are clear, and I like the idea of measuring the topological change of graphs in the spectral space.
2. The experiment results are extensive. I appreciate the authors not only compare many baselines under the graph contrastive learning setup but also consider the transfer learning and adversarial robustness problems.

#### Weaknesses
1. I like the idea of measuring the augmentation in the spectral space. The paper proposes a working algorithm, SPAN, by maximizing the L2 distance of eigenvalues and specifically modeling the edge modification/perturbation. But I think some important or highly-related questions in this direction are not sufficiently discussed and explored in this paper.
    1. One important problem is why maximizing the perturbation (in terms of L2 distance of eigenvalues) can still/always generate valid views of the original graph? Is it necessary to require the $\Delta$ matrix to be sparse (i.e., not too many 1's), and thus the perturbed graph is not too far from the original one? If the perturbed graphs (i.e., the two views generated by SPAN) are too far away from the original one (in terms of the L2 distance of eigenvalues), what topological information is still preserved, and are they still label preserving?
    2. Due to complexity constraints, the authors propose to only compute the $K$ largest and smallest eigenvalues. The authors claim they are the most informative ones, but is there any reference or theory supporting this claim? From my experience, I think it is likely for the real-world graph to have many eigenvalues (of the symmetric normalized Laplacian) equal or close to 0 and 2, but the number of eigenvalues around 1 is relatively small. In this regard, if the chosen $K$ is much smaller than $n$, would it be likely the eigenvalues used to optimize the augmented graph is always equal or close to 0 and 2?
    3. Since this paper only considered the L2 distance between eigenvalues. Thus we are forced to compare graphs of the same size. And only edge perturbation is considered. However, I think we can generalize to consider, e.g., a distributional divergence or two-sample distance, like empirical 1-Wasserstein distance, and generalize the framework to augmentations that also change the number of nodes (e.g., node dropping, subgraph cropping). If this direction is considered out of the scope of this current paper, then the contribution of this paper is somehow limited to a specific type of topological change on graphs and also kind of limit the overall contribution.

**Summary Of The Paper:**

This paper tackles the problem of how to design structural/topological augmentations for graphs, which can be used by graph contrastive learning. The authors aim to find a principled way for topology augmentations by exploring the invariance of graphs from the spectral perspective. This paper proposes to generate topological augmentations guided by maximizing the change in the spectral domain. Experiments on both graph- and node-level tasks demonstrate the
effectiveness of the proposed method in graph self-supervised learning.

**Summary Of The Review:**

Overall I recommend rejection for the current manuscript. My major concern is that several critical questions of the proposed spectral-augmentation approach are not discussed or explored, which limits the contribution of the paper and leaves confusion to the community. I would expect there is also some room to improve the experimental results further if the authors can rethink those important questions of spectral augmentation.

---

> ### Author Response · Authors · 2022-11-14
> **Response to Reviewer fWm4 (part 2/2)**
>
> **[Q4]**: Since it is likely for the real-world graph to have many eigenvalues (of the symmetric normalized Laplacian) equal or close to 0 and 2, would it be likely the chosen $K$ eigenvalues are always equal or close to 0 and 2?
>
> **[A4]**: To show the coverage of $K$ lowest and highest eigenvalues, we first include the eigenvalue distribution on real-world graphs in **Figure 4 in the updated paper**. We can observe that $K=1000$ can cover a moderate range of the eigenvalues, and setting larger $K=5000$ can well cover most of the eigenvalues. Therefore even if we only choose these eigenvalues, we can already achieve empirically satisfactory performance indicated by Figure 3.
>
> The effectiveness of optimizing these eigenvalues can also be found in Appendix D.7, where we observe that the norm of $K$ smallest and largest eigenvalues can be enlarged or reduced as optimization progresses.  Meanwhile, we argue that though the optimization is conducted based on the selected eigenvalues, the whole eigensystem will also be influenced. For example, when we maximize the norm of these eigenvalues, the magnitude of other eigenvalues will be affected as well.
>
> **[Q5]**: We should be able to generalize the framework to augmentations with changing node number (e.g., node dropping, subgraph cropping), and generalize the distance metric to compare graphs with different sizes (e.g., distributional divergence or two-sample distance, like empirical 1-Wasserstein distance).
>
> **[A5]**: We thank the reviewer for suggesting the interesting idea of extending our proposed principle to allow other types of topology augmentations. We want to first clarify that the contribution of this work is to demonstrate a new direction of guiding augmentation via graph spectrum, and we demonstrated its potential using edge augmentation, as it is the most widely adopted augmentation in graph contrastive learning (e.g. in GRACE, GCA, BGRL, GBT, AD-GCL) due to its simplicity.
>
> Following the suggestion from the reviewer, we also tried to extend the principle to **node dropping** augmentation. We design a soft node dropping scheme, which assigns a dropping probability to each node. Different from the edge perturbation scheme, node dropping is sampled from a Bernoulli distribution $\mathcal{B}(\mathbf{p})$, where $\mathbf{p}\in [ 0, 1]^n$. We can sample a node dropping vector $\mathbf{d}\in${0, 1}$^n$, where $d_{i}\sim\mathcal{B}(\mathbf{p})$ indicates whether to drop the node $i$, and the node is dropped if $\mathbf{d}_i=1$.
>
> Dropping a node is equal to removing all the edges connected to this node. Therefore, we can extend the operation of node dropping to edge removal. The node dropping probability $\mathbf{p}$ implies the following edge removing probability matrix $\mathbf{P}$: $\mathbf{P}=\frac{\mathbf{p}\cdot \mathbf{1}^{\top} + (\mathbf{p}\cdot \mathbf{1}^{\top})^{\top}}{2}$, where $\mathbf{1}$ is an all-one vector with dimension $n$.
>
> The node dropping based augmentation scheme is then obtained by: $T(\mathbf{A})=\mathbf{A} + (-\mathbf{A}) \circ\mathbf{P}$, where $\circ$ is an element-wise product. We can replace the edge perturbation scheme $\mathbf{A}+\mathbf{C}\circ \mathbf{\Delta}$ with the node dropping scheme, and following Eq. (5) to optimize $\mathbf{p}$. An augmented view is a sample from the augmentation scheme to drop nodes following the optimized probability $\mathbf{p}$.
>
> As we use soft dropping, the scheme can still be optimized via the L2 distance of eigenvalues. Using distribution divergence to measure the spectral change, suggested by the reviewer, is a very interesting idea which can further enlarge the scope of existing graph augmentations and trigger more applications of our proposed principle.
>
> We empirically compared the spectrum guided node dropping augmentation with uniformly random node dropping strategy. The following table reports their prediction accuracy on four graph classification datasets.
>
> |                                | Biochemical |  Molecules                | Social  Networks|                |
> |--------------------------------|-----------------------|------------------|------------------|------------------|
> |                                | NCI1                  | PROTEINS         | COLLAB           | IMDB-M           |
> | Uniformly random node dropping | 69.27 $\pm$ 0.86      | 73.40 $\pm$ 0.74 | 75.19 $\pm$ 0.67 | 53.04 $\pm$ 0.63 |
> | Spectral guided node dropping  | 70.96 $\pm$ 0.77      | 74.51 $\pm$ 0.58 | 75.65 $\pm$ 0.55 | 53.77 $\pm$ 0.61 |
>
> We can still observe that the spectral guided node dropping augmentation achieves better performance, which demonstrates the applicability of our proposed principle on both edge and node augmentation. We have added the node dropping version of our method and discussed possible future extensions in **Appendix E in the updated paper**.

---

> ### Author Response · Authors · 2022-11-14
> **Response to Reviewer fWm4 (part 1/2)**
>
> We thank the reviewer for the constructive suggestions to clarify the arguments in our work and the interesting ideas of generalizing our proposed principle to other augmentations.
>
> **[Q1]**. Why maximizing the perturbation (in terms of L2 distance of eigenvalues) can still/always generate valid views of the original graph? Is it necessary to require the $\mathbf{\Delta}$ matrix to be sparse (i.e., not too many 1's), and thus the perturbed graph is not too far from the original one?
>
> **[A1]**: Thanks for pointing out the place that unfortunately caused the misunderstanding. We should clarify two different concepts: the **spectral change** evaluated by the L2 distance of eigenvalues is different from the **perturbation intensity** controlled by the constraint $\mathcal{S}$ in Eq. (3). The perturbation intensity $\mathcal{S}$ controls how many edges will be perturbed in expectation, a.k.a. the perturbation budget $\epsilon$, which is a hyper-parameter to be finetuned based on different datasets. The influence of the perturbation budget $\epsilon$ is provided in our answer to CQ2 in the general response to all reviewers. Therefore, it is necessary to require a proper edge perturbation budget, such that the perturbed graphs in both views will not be too far away from the original graph. With such a budget constraint, the obtained probability matrix $\mathbf{\Delta} \in[0,1]^{n\times n}$ serving as the Bernoulli parameters ensures that in expectation, a sampled edge perturbation matrix $\mathbf{E}\in${0, 1}$^{n\times n}$ allows to flip $\epsilon$ edges in both views.
>
> The spectral change is subject to the perturbation budget, instead of allocating the budget uniformly to all edges, our goal is to find the perturbation which can identify a limited number of edges that influence the graph spectrum the most (e.g., to maximize or minimize spectral changes).
>
> **[Q2]**: If the perturbed graphs are too far away from the original one (in terms of the L2 distance of eigenvalues), what topological information is still preserved, and are they still label preserving?
>
> **[A2]**: Following our previous answer, we clarify that under a given perturbation budget $\epsilon$ the perturbed graphs will not be too far away from the original graph. Maximizing the spectral change can best allocate the budget to perturb the edges that introduce a larger variety to spectral invariance.
>
> Since the graph spectrum captures many important structural information as discussed in Appendix A, by maintaining the edges that are stable to the graph spectrum, we capture structural invariance by preserving structural properties which are robust to small edge perturbations.
>
> Here we use the clustering property as an example to intuitively illustrate how the graph with a larger or smaller spectrum looks like. From the perspective of node clustering, as shown in Appendix D.6, increasing the spectrum norm leads to removing edges that bridge different clusters and thus the clustering effect becomes more obvious; on the other hand, decreasing the spectrum norm tends to add edges connecting different clusters, which blurs the clustering effect. The key effect of contrasting these two opposite augmentations is that the information related to these edges (which can easily cause structural change and thus are spurious) is eliminated and disentangled from the learned representations. Since the clustering property is closely related to many node classification tasks (e.g. community detection), preserving the clustering property is also label preserving. Similar observations can be found in the molecular MUTAG dataset in Appendix D.6: the edges between atom groups are perturbed to preserve key chemical groups ($NO_2$) which are usually mutagenic (label preserving).
>
> **[Q3]**: Is there any reference or theory supporting the claim that “the $K$ largest and smallest eigenvalues are the most informative ones”?
>
> **[A3]**: Please find our answer to CQ3 in the general response to all reviewers.

---

> ### Author Response · Authors · 2022-11-18
> **We hope the reviewer find our response helpful**
>
> We appreciate the reviewer for providing many insightful feedbacks, which greatly help us improve the clarify and scope of our paper. We tried to provide thorough response and additional results on node dropping augmentation. We sincerely hope the reviewer find our response useful, and update the scores if your concerns have been resolved.  We are also open to further discussion if there are further questions.

---

### Official Review · Reviewer_SvG5 · 2022-11-06

**Confidence:** 3
**Correctness:** 3
**Technical Novelty And Significance:** 3
**Empirical Novelty And Significance:** 3
**Recommendation:** 8

**Clarity, Quality, Novelty And Reproducibility:**

It is novel and interesting the structural and spectral invariance proposed in this paper. It is clear and easy to understand.

**Strength And Weaknesses:**

Strength:
This paper combines structural invariance with spectral analysis, which is interesting and reasonable for graph contrastive learning.

Weaknesses:
The sensitive edges that can change the graph property should be emphasized when considering the structural invariance. While the spectral augmentations are designed based on the whole graphs, and the edge perturbation is designed for each edge, i.e., the edge flipping is independent.
Then, does this mean that the flipped edges are all sensitive when learning structural invariance? How to evaluate these edges independently? Besides, it would be better to show the influence of different $\epsilon$ in Eq.(3).

**Summary Of The Paper:**

This paper proposed one spectral augmentation scheme for graph contrastive learning methods. It aims to preserve the spectral invariance which is related to the large changes in the graph spectrum. The experiments demonstrate the effectiveness of the proposed method.



**Summary Of The Review:**

This paper proposed one spectral augmentation scheme based on the structural invariance. Extensive experiments demonstrate the effectiveness of the proposed method. There exist one question when evaluate the flipped edges as mentioned before.

---

> ### Author Response · Authors · 2022-11-14
> **Response to Reviewer SvG5**
>
> We thank the reviewer for the positive comments on our work, and the constructive questions to help further clarify our design.
>
> **[Q1]**: The sensitive edges that can change the graph property should be emphasized when considering the structural invariance. While the spectral augmentations are designed based on the whole graphs, and the edge perturbation is designed for each edge, i.e., the edge flipping is independent. Then does this mean that the flipped edges are all sensitive when learning structural invariance? How to evaluate these edges independently?
>
> **[A1]**: Thanks for pointing out the place that could cause unnecessary confusion. We want to first clarify that the edge perturbation scheme is **NOT** optimized independently across edges. We treat the edge augmentation problem as a combinatorial optimization problem, where a set of the most sensitive edges should be perturbed. Since this combinatorial problem is hard to directly optimize, instead of finding a discrete set of edges, we use a probability matrix $\mathbf{\Delta} \in[0, 1]^{n\times n}$ to estimate and quantify the sensitivity of all edges in influencing the graph spectrum, and these probabilities are jointly optimized. After obtaining the probability matrix, we sample edge perturbations from the resulting Bernoulli distributions. We have further clarified the design in **Section 5 in our updated paper**.
>
> **[Q2]**: It would be better to show the influence of different $\epsilon$ in Eq.(3).
>
> **[A2]**: Please find our answer to CQ2 in the general response to all reviewers.

---

### Author Response · Authors · 2022-11-14
**General response to the reviewers**

We sincerely thank all the reviewers for their thoughtful comments and constructive suggestions, which significantly helped us strengthen our paper. We are encouraged to find that the reviewers appreciate the novelty of the proposed idea (Reviewer SvG5, fWm4, 2d5T, eFML) with valid technical details (Reviewer eFML), extensively conducted experiment (Reviewer SvG5, fWm4, 2d5T, eFML), and clear presentation (Reviewer SvG5, fWm4, 2d5T, eFML). There are also shared comments regarding reproducibility, hyperparameter analysis and selective eigenvalues. We now first provide our answers to these common questions, and endeavor to provide individual responses to each reviewer.

**[CQ1]**. Reproducibility.

**[CA1]**: The implementation of our proposed method can be found via https://anonymous.4open.science/r/spectral-augmentation.

**[CQ2]**: Analysis of the hyperparameter $\epsilon$.

**[CA2]**: The value of $\epsilon$ controls the perturbation strength when generating augmented graphs. A larger value indicates that more edges will be dropped/added. Specifically, the optimized scheme $\mathbf{\Delta}_1, \mathbf{\Delta}_2$ constrained by $\epsilon$ will in expectation perturb $\epsilon = \sigma_e \times m$ edges in the augmented views, where $m$ is the total number of edges in the input graph and $\sigma_e$ is the perturbation rate. To analyze the effect of perturbation strength $\epsilon$, we tuned $\sigma_e=\epsilon/m=${0.1,0.2,$\dots$, 0.9} in each dataset, and compare the proposed spectral augmentation with uniformly random edge augmentation on the same GCL instantiation shown in Figure 2. The performance comparison under unsupervised node classification task is provided in **Figure 5, Appendix D.4 in the updated paper**.

From Figure 5, we can observe that the performance of spectral augmentation in general is less sensitive to the hyperparameter perturbation strength, compared with the uniformly random augmentation. This demonstrates that our proposed augmentation assigned larger perturbation probability to the most sensitive edges, without influencing the stable ones too much, and thus it can better preserve structural invariance.

**[CQ3]**: The rationale behind using $K$ largest and smallest eigenvalues should be provided.

**[CA3]**: To reduce the complexity of computing the full graph spectrum via eigendecomposition, we focused on $K$ selective eigenvalues using the Lanczos algorithm. The reasons for selecting $K$ largest and lowest eigenvalues are two-fold:
1. The smallest and largest eigenvalues are most informative to analyze important properties of graphs and GNNs, as discussed in Appendix A. For example, the smallest eigenvalues reflect the connectivity and diameter of a graph [1], and the generalization gap of GNN is closely related to the largest eigenvalue [2].
2. The smallest and largest eigenvalues play different roles in the learning of modern GNNs. From the graph signal processing perspective, the low-frequency components usually carry smoothly varying signals, encouraging neighbor nodes to share similar values, while the high-frequency components carry sharply varying signals across edges, encouraging diversification among neighboring nodes [3]. Recent works [4, 5] reveal that both low and high eigenvalues are important for better graph representation learning.

Based on these reasons, we select both low and high eigenvalues to capture the invariance in both sharply and smoothly varying signals in the augmentation. We elaborate the design of selective eigenvalues in **Section 5 in the updated paper**.

[1] Chung, Fan RK. Spectral graph theory. Vol. 92. American Mathematical Soc., 1997.

[2] Verma, Saurabh, and Zhi-Li Zhang. "Stability and generalization of graph convolutional neural networks." Proceedings of the 25th ACM SIGKDD International Conference on Knowledge Discovery & Data Mining. 2019.

[3] Chang, Heng, et al. "Spectral graph attention network with fast eigen-approximation." Proceedings of the 30th ACM International Conference on Information & Knowledge Management. 2021.

[4] Luan, Sitao, et al. "Is Heterophily A Real Nightmare For Graph Neural Networks To Do Node Classification?." arXiv preprint arXiv:2109.05641 (2021).

[5] Bo, Deyu, et al. "Beyond low-frequency information in graph convolutional networks." Proceedings of the AAAI Conference on Artificial Intelligence. Vol. 35. No. 5. 2021.

---

### Author Response · Authors · 2022-11-17
**A gentle reminder to the reviewers**

Dear reviewers,

Thank you again for your valuable time and thoughtful comments. We have provided thorough responses and additional results. As we are approaching the end of the discussion stage, we would appreciate it if you could read our responses and update the scores if your concerns have been addressed. We are more than happy to further discuss any concerns that you find not fully addressed. Thank you.

Best regards,

Authors

---

### Decision · Program_Chairs · 2023-01-20

**Decision:**

Accept: notable-top-25%

**Justification For Why Not Higher Score:**

This paper does not show a fundamental breakthrough, and it focuses on a specific learning method on graphs.

**Justification For Why Not Lower Score:**

Design GCL method from the spectral perspective is interesting.

**Metareview: Summary, Strengths And Weaknesses:**

This paper proposes a new graph contrastive learning (GCL) method based on spectral perturbation. The designed method is motivated to capture invariance information from graph. Then, a spectral augmentation technique is proposed which can be combined with existing GCL framework. Extensive experiments have been done to show the effectiveness of the proposed method.

Strength
- Authors motivate new design principle of GCL from an interesting perspective.
- Experiments are extensive.

Weakness
- The concept of invariance is not clearly stated (fixed in the revision).
- The connection between structural and spectral invariance is not clearly motivated (fixed in the revision).

**Note From Pc:**

if the above contains the word "oral" or "spotlight" please see: "oral" presentation means -> notable-top-5% and "spotlight" means -> notable-top-25%. As stated in our emails, we are disassociating presentation type from AC recommendations